# The surface longwave cloud radiative effect derived from space lidar observations

**Assia Arouf**[1], **Hélène Chepfer**[1], **Thibault Vaillant de Guélis**[2,3], **Marjolaine Chiriaco**[4], **Matthew D. Shupe**[5,6], **Rodrigo Guzman**[1], **Artem Feofilov**[1], **Patrick Raberanto**[7], **Tristan S. L'Ecuyer**[8], **Seiji Kato**[3], and **Michael R. Gallagher**[5,6]

[1]LMD/IPSL, Sorbonne Université, École Polytechnique, Institut Polytechnique de Paris,
ENS, PSL Université, CNRS, Palaiseau, France
[2]Science Systems and Applications, Inc., Hampton, Virginia, USA
[3]NASA Langley Research Center, Hampton, Virginia, USA
[4]LATMOS/IPSL, UVSQ, Université Paris-Saclay, Sorbonne Université, CNRS, 78280, Guyancourt, France
[5]Cooperative Institute for Research in Environmental Sciences, University of Colorado, Boulder, Colorado, USA
[6]NOAA Physical Sciences Laboratory, Boulder, Colorado, USA
[7]LMD/IPSL, CNRS, Sorbonne Université, École Polytechnique, Institut Polytechnique de Paris,
ENS, PSL Université, Palaiseau, France
[8]Department of Atmospheric and Oceanic Sciences, University of Wisconsin-Madison, Madison, USA

**Correspondence:** Assia Arouf (assia.arouf@lmd.ipsl.fr)

**Abstract.** Clouds warm the surface in the longwave (LW), and this warming effect can be quantified through the surface LW cloud radiative effect (CRE). The global surface LW CRE has been estimated over more than 2 decades using space-based radiometers (2000–2021) and over the 5-year period ending in 2011 using the combination of radar, lidar and space-based radiometers. Previous work comparing these two types of retrievals has shown that the radiometer-based cloud amount has some bias over icy surfaces. Here we propose new estimates of the global surface LW CRE from space-based lidar observations over the 2008–2020 time period. We show from 1D atmospheric column radiative transfer calculations that surface LW CRE linearly decreases with increasing cloud altitude. These computations allow us to establish simple parameterizations between surface LW CRE and five cloud properties that are well observed by the Cloud-Aerosol Lidar and Infrared Pathfinder Satellite Observations (CALIPSO) space-based lidar: opaque cloud cover and altitude and thin cloud cover, altitude, and emissivity. We evaluate this new surface LWCRE–LIDAR product by comparing it to existing satellite-derived products globally on instantaneous collocated data at footprint scale and on global averages as well as to ground-based observations at specific locations. This evaluation shows good correlations between this new product and other datasets. Our estimate appears to be an improvement over others as it appropriately captures the annual variability of the surface LW CRE over bright polar surfaces and it provides a dataset more than 13 years long.

## 1 Introduction

Small changes in the surface irradiance may lead to large climatological responses (Chylek et al., 2007; Kwok and Untersteiner, 2011). Therefore, quantifying irradiance at the Earth's surface is a useful step to better understand the climate system. Clouds exert a very important effect on the energy balance at the surface of the Earth through their effects on shortwave (SW) and longwave (LW) radiation. They radiatively warm the surface in the LW domain because they absorb upward LW radiation that would otherwise escape the Earth system and re-emit it back towards the surface. They cool the surface in the SW domain because they reflect solar radiation back to space that would otherwise partly be

absorbed by the surface. These effects are usually quantified using the surface cloud radiative effect (CRE), defined as the change in the SW and LW radiation reaching the surface induced by the presence of clouds. Globally, clouds radiatively cool the Earth's surface by $20\,\mathrm{W\,m^{-2}}$ according to Kato et al. (2018) and by $25\,\mathrm{W\,m^{-2}}$ according to L'Ecuyer et al. (2019), where the (negative) surface SW CRE cooling is 2 times larger in magnitude than the (positive) surface LW CRE warming. Nevertheless, in some specific regions, like at high latitudes or over the tropical ocean below persistent stratocumulus clouds, the surface LW CRE warming can be larger than the surface SW CRE cooling, so that the clouds exert a net radiative warming of the surface.

As an example, SW effects vanish in the winter-hemisphere polar regions, leading to positive net CRE as LW effects dominate (Henderson et al., 2013). While climate warming in the Arctic is already visible with the sea ice melting (Stroeve et al., 2012), previous works showed that clouds may exert some control on future Arctic climate trajectories (Kay et al., 2012), because they play a primary role in regulating the surface energy balance (Ramanathan et al., 1989; Curry et al., 1996; Shupe and Intrieri, 2004), which influences the surface melting (van den Broeke et al., 2009). Specifically, over Greenland, van Tricht et al. (2016) showed that clouds increase the radiative fluxes into the surface and could therefore modulate the Greenland ice sheet mass balance (van Tricht et al., 2016; Hofer et al., 2017), which is a large contributor to global sea-level rise (Shepherd et al., 2012; IPCC, 2022). At the southern high latitudes, clouds likely exert an important role in the surface energy budget of Antarctica (Shepherd et al., 2012; Kopp et al., 2016), but their radiative impact in this region remains largely unexplored (Scott et al., 2017) in spite of the fact that Antarctica contains the largest reservoir of ice on Earth. King et al. (2015) showed large errors in Antarctic surface energy budget and surface melting rates in models and underlined the importance of improving observations of cloud radiative properties in this region.

Acquaotta and Fratianni (2014) underlined the current urgent need to develop long-term reliable and high-quality climatic time series in order to better understand, detect, predict and react to global climate variability and change. Given the importance of the surface LW CRE and the need for multiyear time series, it is necessary to get reliable estimates of the surface LW CRE over multiple years everywhere around the globe, including over continents and ice-covered regions. The main motivation for the current work is to derive a 13-year time series of the global surface LW CRE that can be used to better understand the cloud property that has driven the evolution of the surface LW CRE during the last decade (Vaillant de Guélis et al., 2017b; Norris et al., 2016). This is a necessary step towards understanding how clouds might interact with the surface in the future as the climate warms (Lindzen and Choi, 2021). A possible way to observe cloud variability is to combine space radar and space lidar observations (Henderson et al., 2013), because passive sensors often struggle to distinguish clouds from the surface over continents and ice-covered regions. The launch of Cloud-Aerosol Lidar and Infrared Pathfinder Satellite Observations (CALIPSO; Winker et al., 2010) and CloudSat Profiling Radar (CPR; Stephens et al., 2008) in 2006 provided the first opportunity to incorporate information about the global vertical cloud distribution (Henderson et al., 2013) over all surface types and is an important parameter for surface LW CRE estimates from space. As CloudSat experienced a battery anomaly that limited future observations to daytime scenes only in 2011, CALIPSO's global observations collected since 2006 are the main tool for providing information on the cloud vertical distribution over more than a decade. Therefore, we retrieve the surface LW CRE from space lidar alone over 13 years.

Section 2 presents the satellite and ground-based data used in this study. In Sect. 3, we present the method followed to retrieve the surface LW CRE from radiative transfer computations. In Sect. 4, we present the radiative-transfer-based statistical regressions tying the surface LW CRE to cloud altitude and emissivity. In Sect. 5, we present the new surface LW CRE retrieved from the analytical relationships and CALIPSO space-based lidar observations (cloud cover, cloud altitude, and cloud opacity). In Sect. 6, we evaluate this new surface LW CRE product against ground-based observations. In Sect. 7, we evaluate it at footprint scale and at $2° \times 2°$ gridded scale against existing independent surface LW CRE satellite-derived products. In Sect. 8, we discuss the limit of the new surface LW CRE product. Section 9 summarizes the main results and perspectives of this work.

## 2   Data

This section describes the CALIPSO cloud observations used to retrieve the surface LW CRE and the independent space-based and ground-based datasets used to evaluate it.

### 2.1   Cloud observations from CALIPSO–GOCCP–OPAQ

We use cloud properties from the GCM Oriented CALIPSO Cloud Product (GOCCP v3.1.2; Chepfer et al., 2010; Cesana et al., 2012; Guzman et al., 2017) over the period 2008–2020. We do not use data collected between 2006 and 2007 because the laser tilted off nadir in November 2007, which introduced some change in the CALIPSO signal. In this product (hereafter, CALIPSO–GOCCP), lidar profiles are classified into three types: clear-sky profile when no cloud is detected, thin cloud profile when one or several cloud layers and a surface echo are detected, and opaque cloud profile when one or several cloud layers are detected but no surface echo is detected. Surface echo is not detected typically when the profile contains a cloud with visible optical depth > 3–5 depend-

ing on the cloud microphysical properties. The cloud base height corresponds to the lowest cloud layer detected. From this classification, five fundamental cloud properties for CRE studies are derived.

- $C_{\text{Opaque}}$: the opaque cloud cover, i.e., the number of opaque cloud profiles divided by the total number of profiles within a $2° \times 2°$ latitude–longitude grid box.

- $Z_{T_{\text{Opaque}}}$: the altitude of opaque cloud, i.e., the average between the altitude of the highest cloud layer in the profile ($Z_{\text{Top}}$) and the altitude of the layer where the lidar beam is fully attenuated ($Z_{\text{FA}}$), is computed for each profile; a schematic illustrating these altitudes is presented in Fig. 1. Then the gridded $Z_{T_{\text{Opaque}}}$ is the average value of all the $Z_{T_{\text{Opaque}}}$ profiles within a grid box.

- $C_{\text{Thin}}$: the thin cloud cover, i.e., the number of thin cloud profiles divided by the total number of profiles within a grid box.

- $Z_{T_{\text{Thin}}}$: the altitude of thin cloud, i.e., the average between the altitude of the highest cloud layer in the profile ($Z_{\text{Top}}$) and the altitude of the lowest cloud layer ($Z_{\text{Base}}$), is computed for each profile; a schematic illustrating these altitudes is presented in Fig. 1. Then, the gridded $Z_{T_{\text{Thin}}}$ is the average value of all the $Z_{T_{\text{Thin}}}$ profiles within a grid box.

- $\varepsilon_{\text{Thin}}$: the thin cloud emissivity, derived from the space lidar retrieval of the thin cloud visible optical depth $\tau_{\text{Thin}}^{\text{VIS}}$ from which we estimate the thin cloud LW optical depth $\tau_{\text{Thin}}^{\text{LW}}$, which is approximately half of $\tau_{\text{Thin}}^{\text{VIS}}$ (Garnier et al., 2015). The relationship $\varepsilon_{\text{Thin}} = 1 - e^{-\tau_{\text{Thin}}^{\text{LW}}}$ (e.g., Vaillant de Guélis et al., 2017a) is computed for each profile and then averaged over all the values within a grid box.

Figure 1 presents the altitudes of interest of an opaque cloud and a thin cloud seen from a downward space-based lidar beam and from an upward ground-based lidar beam. A thin cloud (Fig. 1a) is characterized by three altitudes: $Z_{\text{Top}}$, $Z_{\text{Base}}$ and $Z_{T_{\text{Thin}}}$, which is the average value of the previous two. For an ideal case, these three altitudes are the same when observed from a space-based lidar or a ground-based lidar.

An opaque cloud (Fig. 1b) is characterized by three altitudes. When the lidar is based on the ground, we measure the altitude of the lowest cloud layer ($Z_{\text{Base}}$), the altitude where the lidar beam is fully attenuated ($Z_{\text{FA-G}}$), and $Z_{T_{\text{Opaque-G}}}$, which is the average of the two. When the lidar is onboard a satellite, we measure the highest cloud layer ($Z_{\text{Top}}$), the altitude where the lidar beam is fully attenuated ($Z_{\text{FA}}$), and the average of the two ($Z_{T_{\text{Opaque}}}$).

Figure 2 illustrates the mean $2° \times 2°$ latitude–longitude gridded values of these five variables over the period 2008–2020. At global scale, opaque clouds are more numerous (42 %; Fig. 2a) than thin clouds (25 %; Fig. 2b) in

CALIPSO–GOCCP v3.1.2. Note that these numbers are different from CALIPSO–GOCCP v3.1.1 (35 % and 36 %, respectively), where the threshold used to detect surface echo, which influences the identification of opaque clouds, was lower because CALIPSO–GOCCP v3.1.1 (Guzman et al., 2017) was applied only to nighttime data since noise is lower during nighttime than daytime. CALIPSO–GOCCP v3.1.2 is applied to nighttime and daytime observations. As expected, the multiyear, annual mean opaque and thin cloud altitudes (Fig. 2c, d) reach maxima ($> 9$ km) in the presence of deep convective clouds over the warm pool and over tropical continents and minima ($< 3$ km) in subsidence regions such as over stratocumulus along the western coast of continents. The thin cloud emissivity (Fig. 2e) is larger along the intertropical convergence zone (ITCZ), in the continental regions, and around the Antarctic Peninsula.

## 2.2 Surface LW CRE from satellites

In this subsection, we describe the already existing global surface LW CRE datasets derived from satellite measurements, against which we will evaluate our new satellite retrieval.

### 2.2.1 CERES–CCCM

This product combines Clouds Earth's Radiant Energy System (CERES) radiometer observations of top of the atmosphere (TOA) LW fluxes with observations from CloudSat, CALIPSO and MODIS as well as radiative transfer calculations to retrieve the surface LW fluxes in all-sky and clearsky scenes at a resolution of the CERES Single Scanner Footprint (SSF, 20 km diameter). This product contains the surface LW CRE at the CERES SSF footprint and is part of the CALIPSO, CloudSat, CERES, and MODIS Merged Product (CCCM or C3M: Kato et al., 2010). This product stops in 2011 because of the CloudSat battery anomaly. This product (version RelB1) contains the CERES footprints that include the ground track of CALIPSO and CloudSat. TOA LW fluxes are derived from CERES radiance observations using the Edition 2 Aqua angular distribution model (Loeb et al., 2005, 2007). Surface LW fluxes are computed using cloud properties derived from CALIPSO, CloudSat, and MODIS. CALIOP (Cloud-Aerosol LIdar with Orthogonal Polarization)-derived cloud products are extracted from version 3 of CALIPSO VFM, 0.5 kmALay, and 0.5 kmCLay, and 0.5 kmCPro, products (Vaughan et al., 2018), and R-04 CloudSat CLDCLASS (Sassen and Wang, 2008) and CWC-RO (Austin et al., 2009) products. MODIS cloud properties are derived by the CERES MODIS cloud algorithm described in Minnis et al. (2010). Cloud boundaries derived from CALIOP at a 1/3 km resolution and cloud boundaries derived from CPR CloudSat are merged to form cloud vertical profiles by the method described in Kato et al. (2011). These cloud profiles are further merged into CERES foot-

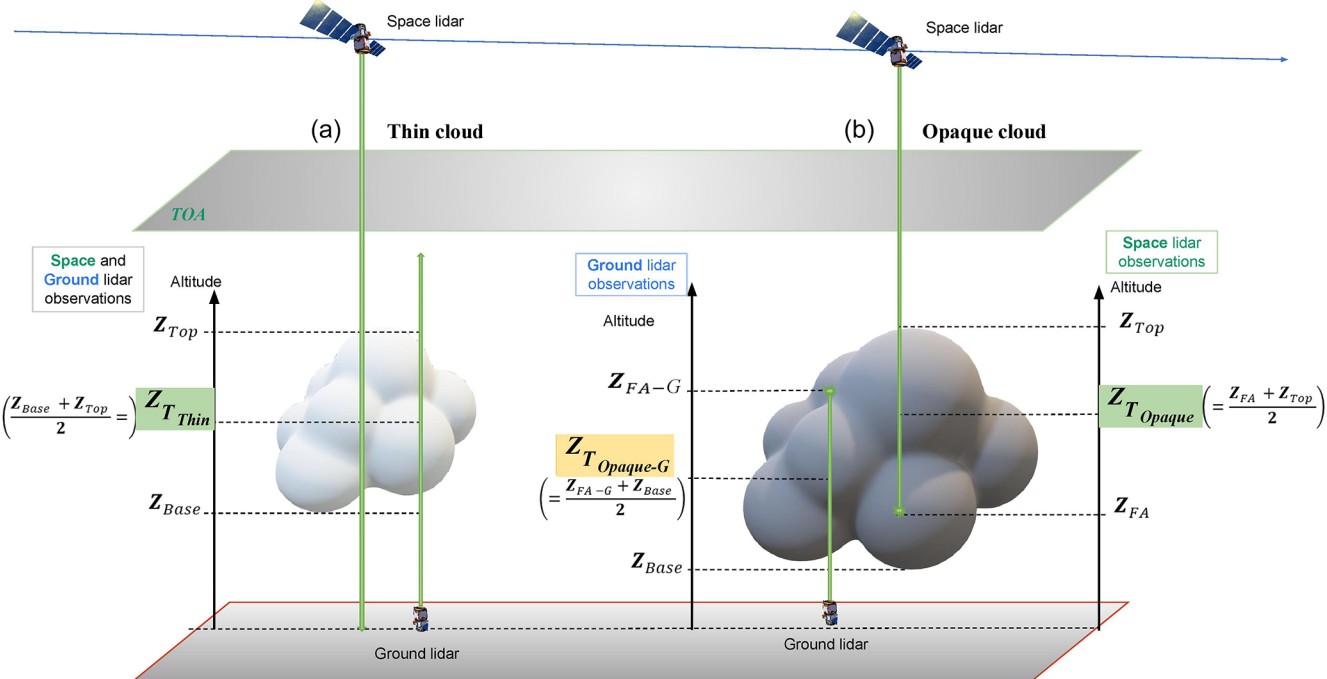

**Figure 1.** Schematic of cloud altitudes seen from space lidar and from a ground-based lidar in an atmospheric column containing thin cloud only **(a)** and opaque cloud only **(b)**. The altitudes used to retrieve the surface LW CRE from CALIPSO–GOCCP are reported in green.

prints, the sizes of which are approximately 20 km. Temperature and humidity profiles used in flux computations are from the Goddard Earth Observing System Data Assimilation System reanalysis (GEOS, Rienecker et al., 2008). GEOS-4 is used from July 2006 through October 2007, and GEOS-5.2 is used from November 2007. Further description of the RelB1 CERES-CCCM product is given in Ham et al. (2017) and Kato et al. (2019).

### 2.2.2 2B-FLXHR-LIDAR

The CloudSat 2B–FLXHR–LIDAR P1_R04 (hereafter, 2BFLX) product combines measurements from CloudSat, CALIPSO, and MODIS to generate estimates of longwave and shortwave fluxes and heating rates throughout the atmosphere (L'Ecuyer et al., 2008; Henderson et al., 2013). The algorithm uses inferred vertical profiles of cloud and precipitation water contents and particle size and temperature and humidity profiles from ECMWF analyses as input to a broadband radiative transfer model. A detailed description of the approach used to reconstruct the atmospheric columns and prescribe surface characteristics as well as a thorough uncertainty assessment is provided in Henderson et al. (2013) and Matus and L'Ecuyer (2017). The surface LW CRE product used here is provided for each CloudSat orbit at the instantaneous footprint scale of 1.8 km and gridded for the comparisons that follow. The dataset currently covers the period August 2006 through April 2011 before CloudSat experienced

a battery anomaly that limited operations to daylight conditions.

The surface fluxes derived from a combination of radar and lidar observations in 2BFLX are less susceptible to uncertainties due to undetected multi-layered clouds and uncertainties in cloud base height than those derived primarily from passive observations (L'Ecuyer et al., 2019; Hang et al., 2019). However, both 2BFLX as well as the LWCRE–LIDAR product are sensitive to retrieval errors and biases introduced by the limited spatial and temporal characteristics of CloudSat and CALIPSO. Sensitivity studies suggest that uncertainties in monthly-mean surface longwave irradiances at 2.5° resolution derived from 2BFLX are $\sim 11\,\mathrm{W\,m^{-2}}$, owing primarily to errors in specifying lower tropospheric temperature and humidity and uncertainty in cloud base height (Henderson et al., 2013).

### 2.3 Surface LW CRE from ground-based sites

As the retrieval of the surface CRE from space observations is not direct, we will evaluate the surface LW CRE retrieved from space against that derived from surface radiation measurements collected directly at ground-based sites. For this purpose, we selected three sites located in different regions.

The first site is located in the Arctic, where constraining radiative transfer is challenging with the limited cloud, available atmospheric temperature and humidity profile observations (Kay et al., 2015) and where the surface CRE may influence the Greenland ice-cap melt (van Trich et al., 2016;

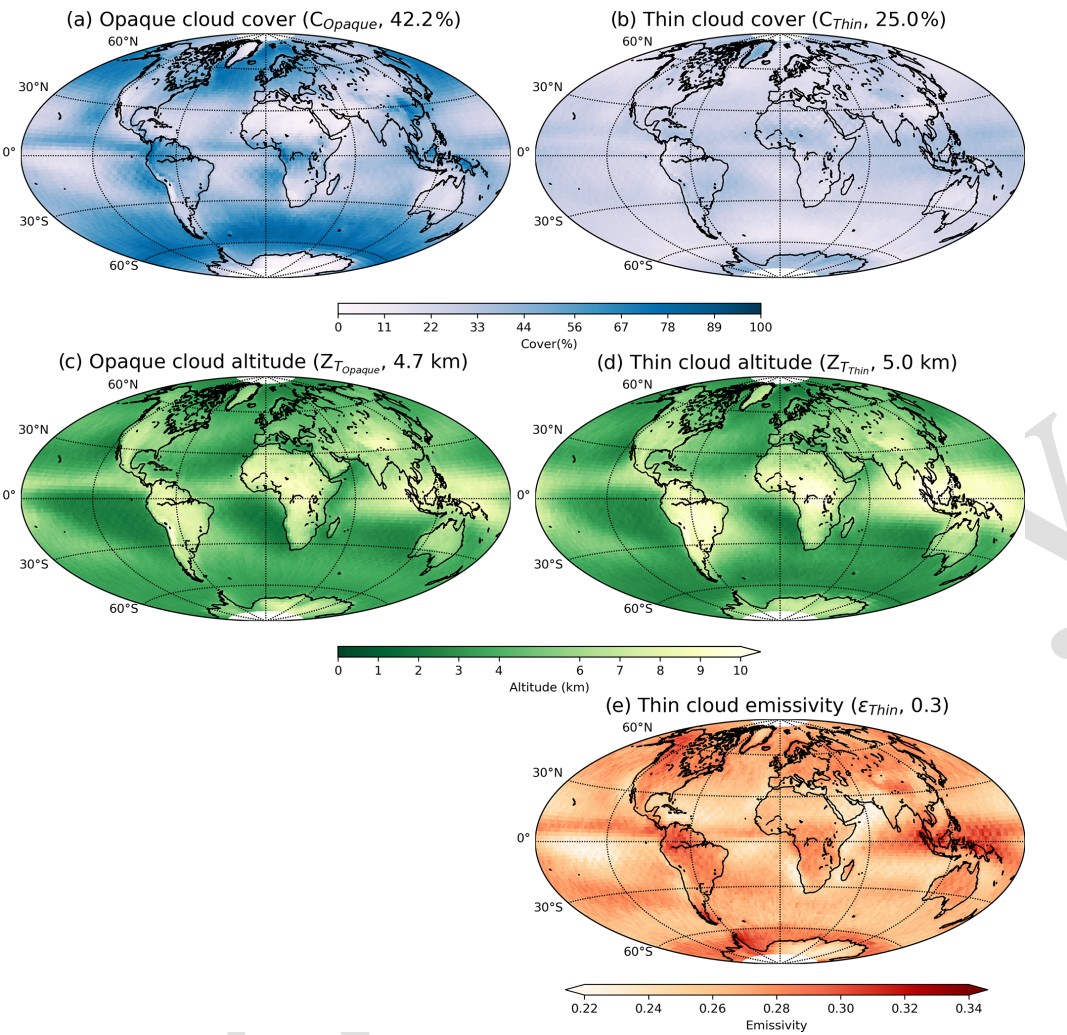

**Figure 2.** Maps of **(a)** opaque cloud cover $C_{\mathrm{Opaque}}$, **(b)** thin cloud cover $C_{\mathrm{Thin}}$, **(c)** opaque cloud altitude $Z_{T_{\mathrm{Opaque}}}$, **(d)** thin cloud altitude $Z_{T_{\mathrm{Thin}}}$ and **(e)** thin cloud emissivity $\varepsilon_{\mathrm{Thin}}$. Global means are reported in parentheses. Build from CALIPSO–GOCCP v3.1.2 over 2008–2020.

Hofer et al., 2017; Shupe et al., 2013). This Summit station (Shupe et al., 2013; Gallagher et al., 2018) is located at the top of the Greenland ice cap (72.6° N–38.5° W) with an elevation of 3250 m. Summit is unique because it is the only place where we have enough observations to make a robust assessment of the surface CRE over Greenland (Lacour et al., 2018). Here, the clear-sky flux is computed using a radiative transfer algorithm with measurements of temperature and humidity profiles (e.g., REFs), while the all-sky flux is measured directly using a pair of upward- and downward-looking broadband pyrgeometers (e.g., Shupe and Intrieri, 2004; Intrieri et al., 2002).

The second site is located at continental mid-latitudes. This Site Instrumental de Recherche par Télédétection Atmosphérique (SIRTA, Haeffelin et al., 2005; Chiriaco et al., 2018) is located in France (48.7° N–2.2° E) with an elevation of 156 m. The data are part of the Baseline Surface Radiation Network (BSRN; Ohmura et al., 1998; Driemel et al., 2018).

At SIRTA, the clear-sky flux is a parameterization made from the surface humidity, the integrated moisture content over the atmospheric column and the air temperature at 2 m. The details are given in Dupont and Haeffelin (2008), and this product has also been used in Rojas et al. (2021). The resulting clear-sky uncertainty is approximately $\pm 5\,\mathrm{W\,m^{-2}}$.

The third site is located in the tropical belt, where clouds influence the global climate and heat transport (Loeb et al., 2016) and where extensive deep convective clouds reach the cold tropical tropopause. Here, the surface LW CRE is small, since much of the surface downward LW radiation originates from emission by the moist near-surface layers of the atmosphere (Prata, 1996). This Kwajalein station (KWA, Roesch et al., 2011), which is also part of the BSRN, is located in the northern Pacific Ocean (8.72° N–167.73° E) with an elevation of 10 m.

Over the three ground-based sites, the radiative flux measurements at the surface are carried out using two Kipp and

Zonen CM22 pyrgeometers, which measure in the spectral range of 4.5–40 μm.

# 3  Method

## 3.1  Approach

Vaillant de Guélis et al. (2017a, b) retrieved the TOA LW CRE from the five CALIPSO–GOCCP cloud properties presented in Fig. 2: the opaque cloud cover, the opaque cloud altitude, the thin cloud cover, the thin cloud altitude, and thin cloud emissivity. In adapting their approach to the surface instead of the TOA, we developed a method to retrieve the surface LW CRE from the same five CALIPSO–GOCCP cloud properties. The method we have developed is based on simple parameterization. This will allow us to, in future work, more easily decompose the temporal variations of the surface LW CRE into several components in order to identify which cloud variables have driven the variations of the surface LW CRE during the last 13 years. The following physical differences exist between the surface and the TOA.

### 3.1.1  Moisture

Moisture within the boundary layer influences the surface LW CRE more than the TOA LW CRE. To take moisture effects into account, we add the surface elevation in the framework of Vaillant de Guélis et al. (2017), and we consider different humidity and temperature profiles at a monthly resolution and for every 2° latitude, differentiating oceans from continents. Compared to the fluxes themselves, small water vapor variability does not affect CRE much, as the equivalent clear-sky contribution is removed from CRE. The surface LW CRE dependence on temperature and humidity profiles is shown in Sect. 4.1, 4.2, and 4.3. The impact on the results of using monthly-mean humidity and temperature profiles will be discussed in Sect. 8.

### 3.1.2  Cloud heights used for the surface LW CRE estimate

In thin cloud situations, the surface LW CRE is influenced by radiation emitted downwards by all cloud layers between the cloud base ($Z_{\text{Base}}$) and the cloud top ($Z_{\text{Top}}$). Therefore, the surface LW CRE depends on the vertical distribution of condensed water between $Z_{\text{Base}}$ and $Z_{\text{Top}}$, which are measured by the lidar. Therefore, we use the average of both ($Z_{T_{\text{Thin}}}$) to estimate the surface LW CRE from lidar observations.

In opaque cloud situations, the surface LW CRE is influenced by radiation emitted by all cloud layers between cloud base ($Z_{\text{Base}}$) and the altitude $Z_{\text{Emisv1}}$, defined as the altitude where the emissivity between $Z_{\text{Base}}$ and $Z_{\text{Emisv1}}$ is close to 1. Cloud layers at altitudes higher than $Z_{\text{Emisv1}}$ do not contribute to the surface LW CRE. Therefore, the surface LW CRE depends on the vertical distribution of condensed water between $Z_{\text{Base}}$ and $Z_{\text{Emisv1}}$.

In those specific cases where the vertical distribution of condensed water is such that $Z_{\text{Base}}$ equals $Z_{\text{Emisv1}}$, meaning that the $Z_{\text{Base}}$ layer (480 m thick) contains enough condensed water to alone make the emissivity close to 1, then the surface LW CRE is driven only by $Z_{\text{Base}}$. In that specific case, the lidar $Z_{\text{FA}}$ should be used to compute the surface LW CRE, and the larger the difference between $Z_{\text{FA}}$ and the actual $Z_{\text{Base}}$, the more the space-based lidar surface LW CRE will be underestimated.

In all other opaque cloud situations, where $Z_{\text{Base}}$ is lower than $Z_{\text{Emisv1}}$, all cloud layers between $Z_{\text{Base}}$ and $Z_{\text{Emisv1}}$ contribute to the surface LW CRE, and the relative weight of each layer depends on the detailed vertical distribution of condensed water between $Z_{\text{Base}}$ and $Z_{\text{Emisv1}}$. In that case, the lidar measures $Z_{\text{Top}}$ and $Z_{\text{FA}}$, and we use the average $Z_{T_{\text{Opaque}}}$, which is the average of $Z_{\text{Top}}$ and $Z_{\text{FA}}$ to estimate the surface LW CRE from lidar observations.

To retrieve the surface LW CRE, we could use $Z_{\text{Base}}$ from CloudSat, but this would limit our time series to 2011 only instead of 2021, and CloudSat is not always optimal for detecting cloud base, in particular if it is a liquid-water cloud. We chose to use what we have access to with CALIPSO: a first option consists in using $Z_{\text{FA}}$, the lowest opaque cloud altitude observable by space lidar ($Z_{\text{FA}} < 3$ km above the surface most of the time, Guzman et al., 2017), which is close to the actual cloud base height except in deep convective towers and some frontal mid-latitude clouds. A second option is to use $Z_{T_{\text{Opaque}}}$, which might represent the altitude of emission of the cloud in some cases. This second option will overestimate the mean altitude of the deep convective towers, where the downward space-based lidar beam attenuates quickly without seeing much of the cloud bottom. The bias will be larger when the cloud base temperature is far from that of $Z_{\text{FA}}$. Moreover, this bias will depend on the opacity of the part of the cloud laying under $Z_{\text{FA}}$ that is not observable by space lidar.

Hereafter, we describe the method with $Z_{T_{\text{Opaque}}}$. Afterwards, we show the results for both option 1 ($Z_{\text{FA}}$) and option 2 ($Z_{T_{\text{Opaque}}}$).

The impact of the results on using these cloud heights will be discussed in Sects. 7 and 8.

## 3.2  Definition of the radiative quantities

In order to get simple notation and because we are only interested in the CRE at the surface in the LW domain in this study, the surface LW CRE will simply be denoted "CRE" in the following equations.

To infer "CRE", the net LW radiative fluxes over all types of scenes ($F_{\text{Allsky}}^{\text{net}}$) may be compared with corresponding fluxes where the influence of clouds has been removed ($F_{\text{Cloudy}-\text{freesky}}^{\text{net}}$). Then, we define the surface LW CRE as fol-

lows:

$$CRE = F_{\text{Allsky}}^{\text{net}} - F_{\text{Cloudy-freesky}}^{\text{net}}. \tag{1}$$

Using downwelling ($\downarrow$) and upwelling ($\uparrow$) fluxes, the surface LW CRE is expressed as follows:

$$CRE = \left( F_{\text{Allsky}}^{\downarrow} - F_{\text{Allsky}}^{\uparrow} \right) - \left( F_{\text{Cloudy-freesky}}^{\downarrow} - F_{\text{Cloudy-freesky}}^{\uparrow} \right). \tag{2}$$

Rearranging the terms on the right-hand side of this equation, we get

$$CRE = \left( F_{\text{Allsky}}^{\downarrow} - F_{\text{Cloudy-freesky}}^{\downarrow} \right) - \left( F_{\text{Allsky}}^{\uparrow} - F_{\text{Cloudy-freesky}}^{\uparrow} \right), \tag{3}$$

which can also be expressed as

$$CRE = CRE^{\downarrow} - CRE^{\uparrow}, \tag{4}$$

where $CRE^{\downarrow}$ represents the surface CRE on the LW downward fluxes and $CRE^{\uparrow}$ the surface CRE on the LW upward fluxes. $CRE^{\uparrow}$ does not exceed $1\,\text{W}\,\text{m}^{-2}$ in the annual global average (Allan, 2011) and in the radiative transfer computations. Therefore, the error in the surface properties plays a minor role.

Nevertheless, in the LW domain, clouds can warm the surface, changing the surface temperature, which is then related to the upwelling LW radiation. This is a subtle but important issue and is dependent to some degree on the surface type (i.e., land surface will warm more than ocean). If "CRE" is determined in a hypothetical way, one could assume that the surface temperature is the same. However, this does not capture the full impact of the clouds. To understand the full impact of the clouds, one would need to consider the adjustments of all other parameters, most importantly the surface temperature. In this study we assume that the surface temperature is the same under clouds and clear skies, consistent with the definition used in previous satellite-derived products (e.g., Kato et al., 2018; L'Ecuyer et al., 2019).

### 3.3 Radiative transfer simulations

We use a radiative transfer code to compute the surface LW CRE due to an opaque cloud ($CRE_{\text{Opaque}}$) or an optically thin cloud ($CRE_{\text{Thin}}$) in an atmospheric column fully overcast by that cloud. In these 1D atmospheric columns, molecules and clouds are evenly distributed within each layer, and each layer is considered infinite and homogeneous. For a single column fully overcast by an opaque cloud, we derived a parameterization between $CRE_{\text{Opaque}}$ and the opaque cloud altitude $Z_{T_{\text{Opaque}}}$ (see Sect. 2.1). For the single column fully overcast by a thin cloud, we derived a parameterization between $CRE_{\text{Thin}}$, the thin cloud altitude $Z_{T_{\text{Thin}}}$ (see Sect. 2.1)

and the thin cloud emissivity $\varepsilon_{\text{Thin}}$, as in Vaillant de Guélis et al. (2017).

The radiative transfer simulations are performed with GAME (Dubuisson et al., 2004). This radiative transfer code computes LW fluxes at 50 different levels with a vertical resolution of 1 km in the first 25 levels. The fluxes are spectrally integrated between 5 and $200\,\mu\text{m}$, consistent with CERES measurements. We prescribe various surface temperatures and the atmospheric profiles of humidity, temperature, ozone and pressure based on ERA-Interim reanalysis (Dee et al., 2011) over oceans and lands for each month and 2° latitude. Humidity and temperature profiles over land for January are presented in Fig. A2 in Appendix A. Figure A3 presents the seasonal and latitudinal behavior of the first layer of the humidity and temperature profiles (from the surface to 1 km above the surface) over ocean and over land. We perform all-sky fluxes through radiative transfer computations for numerous combinations of cloud opacity and vertical distribution. We prescribe the vertical extent of each cloud, the effective size of cloud particles and the infrared optical thickness. For a column fully overcast by an opaque cloud, the cloud is represented by a 1 km-thick cloud layer with an emissivity close to 1 at $Z_{\text{FA–G}}$ ($Z_{\text{Top}}$) above optically uniform cloud layers for different vertical extents with a vertically integrated emissivity equal to 0.8. For a column fully overcast by a thin cloud, the cloud is represented by optically uniform cloud layers with vertically integrated emissivities equal to 0.1, 0.3, 0.5 or 0.7. The cloud top altitude varies according to latitude and can reach 17 km in tropical regions and only 11 km in polar regions. For instance, the cloud top altitude at a latitude of 39° N takes 11 different values ranging between 2 and 13 km, and for each cloud top value, the cloud base altitude takes all possible values between 1 km above the surface and the cloud top altitude minus 1 km. Clear-sky fluxes are defined by recalculating fluxes after removing clouds with the same humidity and temperature profiles.

### 3.4 Retrieval of the surface LW cloud radiative effect from CALIPSO observations and radiative transfer simulations

The surface LW CRE is retrieved from parameterizations derived from radiative transfer simulations that involve five observed CALIPSO–GOCCP cloud properties. Two surface LW CRE datasets are built from the CALIPSO–GOCCP product using this theoretical relationship over the 2008–2020 period, an orbit dataset at the CALIOP footprint resolution of instantaneous cloud property observations and a $2° \times 2°$ gridded dataset of mean cloud properties. For the orbit dataset, each lidar profile contains either an opaque or thin cloud or no cloud, and the surface LW CRE for this last category is zero. For the gridded product, at each grid point, the opaque surface LW CRE is computed from the gridded $Z_{T_{\text{Opaque}}}$ and weighted by the gridded opaque cloud cover $C_{\text{Opaque}}$ in the same way as Vaillant-de-Guélis et al. (2017).

The thin surface LW CRE is computed from the gridded $Z_{T_{\text{Thin}}}$ and gridded $\varepsilon_{\text{Thin}}$ and then weighted by the gridded thin cloud cover $C_{\text{Thin}}$. The total gridded surface LW CRE is the sum of the two.

$$\text{CRE} = \text{CRE}_{\text{Opaque}} + \text{CRE}_{\text{Thin}} \tag{5}$$

In the retrievals, we tested both $Z_{\text{FA}}$ and $Z_{T_{\text{Opaque}}}$ for estimating the mean altitude of opaque clouds (as discussed in Sect. 3.1).

The new product name is "LWCRE–LIDAR–Ed1" for "LW Cloud Radiative Effect derived from space Lidar observations Edition 1", and the acronyms are LWCRE–LIDAR and CRE$_{\text{LIDAR}}$ in this study. This new monthly gridded product is available for the 2008–2020 time period at https://doi.org/10.14768/70d5f4b5-e740-4d4c-b1ec-f6459f7e5563 (Arouf et al., 2022), and Table C1 summarizes the data included in the dataset.

## 4   Modeled CRE sensitivity to cloud properties

This section establishes parameterizations of the surface LW CRE against cloud altitude and emissivity over a single cloudy column using radiative transfer computations (Sect. 4.1). Then it analyzes the sensitivity of the surface LW CRE to the humidity and temperature profiles (Sect. 4.2) and to the surface elevation (Sect. 4.3).

### 4.1   Sensitivity of the CRE to cloud altitude

Figure 3 shows the results of numerous simulations for the opaque cloud column (Fig. 3a) and the thin cloud column (Fig. 3b) for a specific atmospheric state over oceans in January at a latitude of 39° N. CRE$_{\text{Opaque}}$ decreases approximately linearly with opaque cloud altitude at a rate of $6.0\,\text{W}\,\text{m}^{-2}\,\text{km}^{-1}$ in this atmospheric state. This figure shows that the surface LW cloud radiative effect depends mostly on the mean altitude of the cloud and only weakly on the detailed vertical cloud distribution and the cloud bottom altitude. CRE$_{\text{Thin}}$ also decreases linearly with thin cloud altitude, and the rate of decrease depends linearly on the cloud emissivity. The linearity of these relationships is consistent with Ramanathan (1977) and Vaillant de Guélis et al. (2017, 2018). It is an empirical relation derived from radiative transfer calculations that has been verified in the observation at the TOA in Vaillant de Guélis et al. (2017a, 2018). Corti and Peter (2009) also derived an empirical relationship (power laws) from radiative transfer computation. Our linear relationship can be seen as an approximation of the Corti and Peter (2009) power law.

Based on a regression, we obtain the following linear relationships between the surface LW CRE and cloud altitude and emissivity:

$$\text{CRE}_{\text{Opaque}} = C_{\text{Opaque}} \times \big[ a\,(\text{RH}, T) \times Z_{T_{\text{Opaque}}} \\ + b\,(\text{RH}, T) \big], \tag{6}$$

$$\text{CRE}_{\text{Thin}} = C_{\text{Thin}} \times (\varepsilon_{\text{Thin}} + 0.06) \times \big[ a\,(\text{RH}, T) \\ \times Z_{T_{\text{Thin}}} + b\,(\text{RH}, T) \big], \tag{7}$$

where $a(\text{RH},T)\,\text{W}\,\text{m}^{-2}\,\text{km}^{-1}$ and $b(\text{RH},T)\,\text{W}\,\text{m}^{-2}$ are constants whose values depend on the humidity and temperature profiles as discussed hereafter. For the specific case presented in Fig. 3, $a = -6.0\,\text{W}\,\text{m}^{-2}\,\text{km}^{-1}$ and $b = +88.0\,\text{W}\,\text{m}^{-2}$.

### 4.2   Sensitivity of the CRE to humidity and temperature profiles

The temperature and humidity profiles in the first layers of the atmosphere largely vary according to seasons and location as presented in Fig. A3 in Appendix A. Since these are variables that influence the surface LW CRE, their variations must be taken into account in order to retrieve the global surface LW CRE.

As an example, Fig. 4a presents the opaque surface LW CRE for a standard humidity profile and Fig. 4b presents the opaque surface LW CRE for an enhanced humidity profile (shown in Fig. A4). A 10 % change in humidity in the first few kilometers of the tropical atmosphere leads to a surface LW CRE change of $7.7\,\text{W}\,\text{m}^{-2}$ for a cloud at 1 km and by $5\,\text{W}\,\text{m}^{-2}$ for a cloud at 4 km. To capture some variability of humidity and temperature, we have established similar relationships as in Fig. 3 for each month and latitude (every 2°) over land and ocean. As an example, Fig. A1 shows the simulations for cloud columns for an atmospheric state over land in January at a latitude of 39° N (same as Fig. 3 but over land instead of ocean). At this latitude, the amount of humidity is lower over land than ocean, and therefore the LW $F_{\text{Cloudy}-\text{freesky}}^{\text{net}}$ over land is lower and the surface LW CRE would be larger than over the ocean. The surface LW CRE is greater than that over ocean and decreases at a rate $(a(\text{RH},T)\,\text{W}\,\text{m}^{-2}\,\text{km}^{-1})$ of $6.5\,\text{W}\,\text{m}^{-2}\,\text{km}^{-1}$ instead of $6.0\,\text{W}\,\text{m}^{-2}\,\text{km}^{-1}$ over ocean. Figure 5 presents the latitudinal and seasonal behavior of the linear regression coefficients $(a(\text{RH},T)\,\text{W}\,\text{m}^{-2}\,\text{km}^{-1}$ and $b(\text{RH},T)\,\text{W}\,\text{m}^{-2})$. The shape of these coefficients' spatiotemporal variation is influenced by the shape of the seasonal cycle of humidity and temperature in the first layers of the atmosphere (Fig. A3). For instance, the behavior of the intercept $(b(\text{RH},T)\,\text{W}\,\text{m}^{-2})$ over ocean and land (Fig. 5b and d, respectively) is driven by the shape of the humidity amount where the largest humidity amount (in tropical regions) causes the smallest intercept coefficients. The seasonal cycle of the surface LW CRE is more pronounced over land than over ocean because the seasonal cycles of humidity and temperature are more pronounced over land than over ocean due to the heat capacity of the surface (Chepfer et al., 2019).

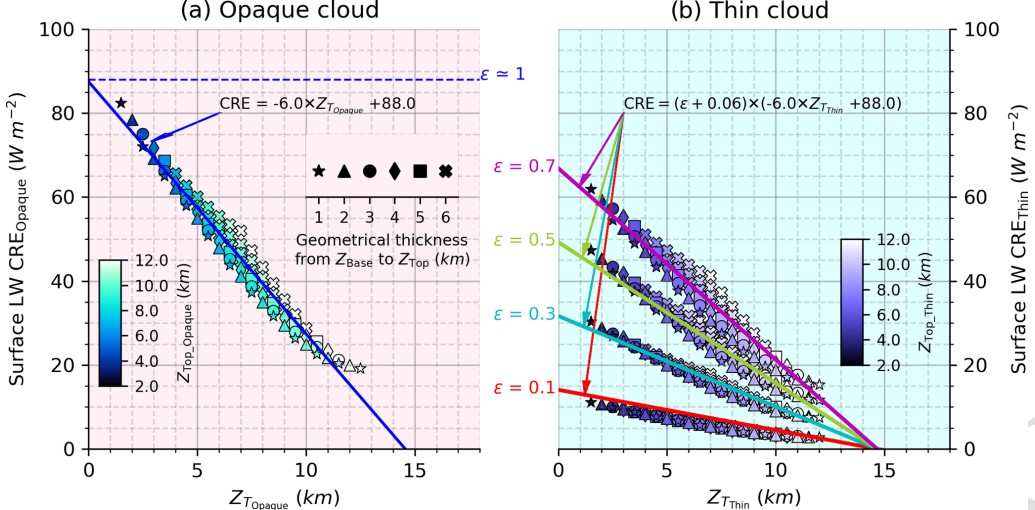

**Figure 3.** Linear relationships derived from 1D radiative transfer computations between the surface LW CRE and the cloud altitude for a single overcast column containing **(a)** an opaque cloud above a thin cloud, both moving in altitude, and **(b)** a thin cloud of emissivity 0.1 (red), 0.3 (cyan), 0.5 (green) and 0.7 (pink). These linear relationships (solid lines) are derived from direct radiative transfer computations (dots). Each dot represents the result of one radiative transfer computation. The color of dots represents the cloud top altitude (2 km, dark, to 13 km, bright) and the shape of dots the geometrical thickness from the cloud base to cloud top (1 km, a star, to 6 km and above, a cross). The atmospheric state is taken from ERA-Interim reanalysis for January at a latitude of 39° N over the ocean. As an example, in plot **(a)** the slope is $-6.0\,\mathrm{W\,m^{-2}\,km^{-1}}$, and the intercept is $88.0\,\mathrm{W\,m^{-2}}$.

## 4.3 Sensitivity of the CRE to surface elevation

In order to take the surface elevation in the simulation into account, we consider the surface temperature to be the temperature of the atmospheric layer located at the same altitude as the surface elevation with respect to sea level, and we discard all layers located between sea level and the altitude of surface elevation. We then performed numerous radiative transfer simulations corresponding to different clouds, as described in Sect. 3.2.

The results presented in Fig. 6 show the sensitivity of the surface LW CRE to the surface elevation over continents in January at 39° N. As the surface elevation increases, the atmosphere is drier, so $F_{\mathrm{Cloudy-freesky}}^{\mathrm{net}}$ decreases and the surface LW CRE increases. The same cloud with the same cloud properties (i.e., same altitude and emissivity) will warm a surface with a high elevation more than a low elevation. For instance, an opaque cloud at an altitude of 5.5 km m.s.l. (mean sea level) will warm a surface at sea level by $\sim 58\,\mathrm{W\,m^{-2}}$ and a surface with an elevation of 4 km m.s.l. by $\sim 102\,\mathrm{W\,m^{-2}}$. These results are consistent with Wang et al. (2019), who found that the surface LW CRE increases over the Summit station in Greenland due to the dry atmosphere at high elevations. We performed radiative transfer simulations for different surface elevations at all latitudes and months (not shown) and used these to retrieve the surface LW CRE from space-based lidar observations over land. Thus, the regression coefficients over land

also depend on surface elevation, with a 100 m resolution ($a(\mathrm{RH},T,\mathrm{SE})\,\mathrm{W\,m^{-2}\,km^{-1}}$, $b(\mathrm{RH},T,\mathrm{SE})\,\mathrm{W\,m^{-2}}$).

## 5 New surface LW cloud radiative effect derived form CALIPSO–GOCCP: LWCRE–LIDAR

### 5.1 Orbit product

Figure 7 (first panel) show the CALIPSO–GOCCP cloud vertical mask (Guzman et al., 2017) for two different parts of an orbit, both in the tropical region. The blue areas over green areas represent the opaque clouds. The blue areas over white areas represent thin clouds. The second line represents the instantaneous surface LWCRE–LIDAR derived from CALIPSO–GOCCP instantaneous cloud properties (opaque cloud altitude, thin cloud altitude and emissivity; $\mathrm{CRE}_{\mathrm{LIDAR}}$), as described in Sect. 3.3. As expected, the surface LWCRE–LIDAR is larger for opaque clouds (Fig. 7a, $\sim 22\,\mathrm{W\,m^{-2}}$) than for thin clouds (Fig. 7b, $\sim 5\,\mathrm{W\,m^{-2}}$) for almost the same atmosphere.

### 5.2 Gridded product

Figure 8a shows the map of the surface LWCRE–LIDAR derived from the CALIPSO–GOCCP product over the 2008–2020 time period.

In annual global means, clouds radiatively warm the surface in the LW domain by $27.0\,\mathrm{W\,m^{-2}}$. $\mathrm{CRE}_{\mathrm{LIDAR}}$ is maximal in the Southern Ocean ($\sim 50$–$65\,\mathrm{W\,m^{-2}}$), where the

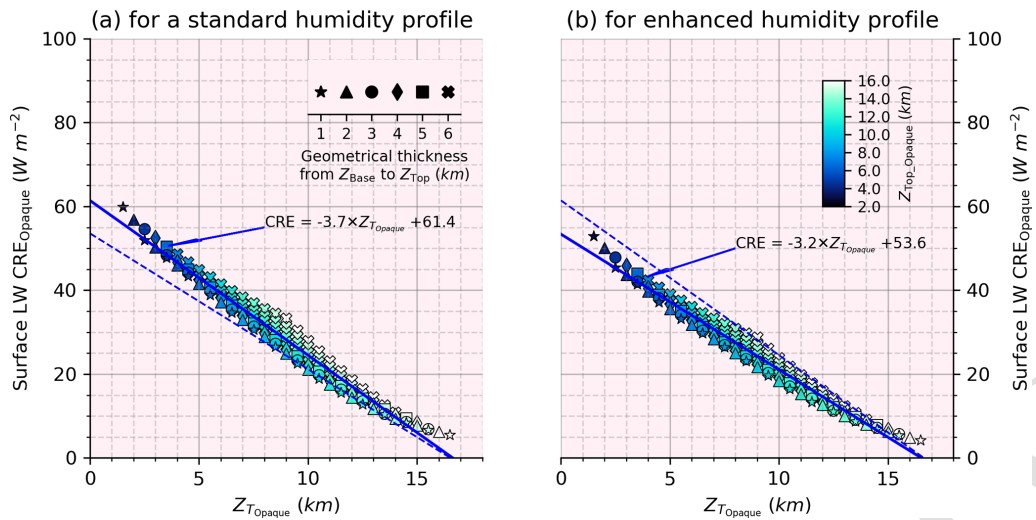

**Figure 4.** Same as Fig. 3a for **(a)** a standard humidity profile and **(b)** an enhanced humidity profile, both in the tropics: [30° S–30° N].

**Figure 5.** Coefficients of the linear relationships derived from 1D radiative transfer computations between the surface LW CRE and the cloud altitude for all latitudes and seasons: **(a)** the slope of the relationships over ocean, **(b)** the intercept of the relationships over ocean, **(c)** the slope of the relationships over land and **(d)** the intercept of the relationships over land.

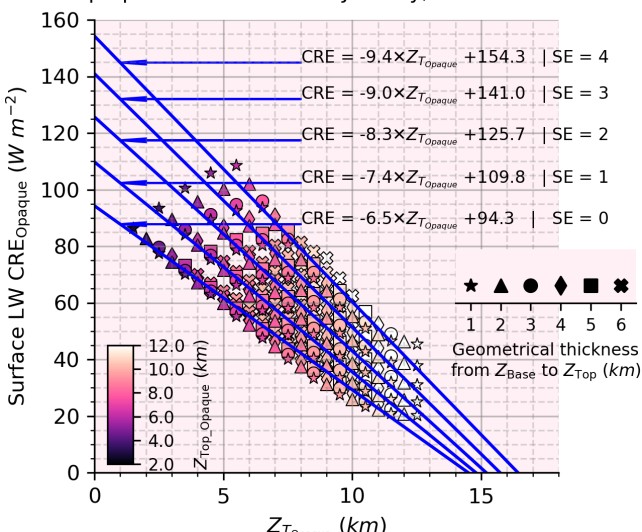

**Figure 6.** Sensitivity of the surface LW opaque CRE to the surface elevation (SE): same as Fig. 3a but over land and for different values of SE: SE = 0 (sea level), SE = 1 km, SE = 2 km, SE = 3 km, and SE = 4 km for January at a latitude of 39° N.

warm opaque low clouds are numerous, as already stated by L'Ecuyer et al. (2019) and Henderson et al. (2013). There are also particularly high values in the North Atlantic ($> \sim 55 \, \text{W m}^{-2}$) observed between Svalbard and Greenland. In the tropics, clouds typically radiatively warm the surface in the LW domain by only $\sim 15 \, \text{W m}^{-2}$. The moist tropical oceanic atmosphere enhances the downward clear-sky fluxes, which decreases the surface LW CRE over these oceans. The maximum tropical $\text{CRE}_{\text{LIDAR}}$ ($\sim 30$ to $\sim 40 \, \text{W m}^{-2}$) is produced by warm opaque low oceanic stratocumulus clouds along the western coast of the continents.

Over continents, the weakest $\text{CRE}_{\text{LIDAR}}$ ($< \sim 5 \, \text{W m}^{-2}$) occurs over the Wadi Abadi basin in the Egyptian desert (25° N, 33° E), a cloud-free region most of the time (80 %). The largest $\text{CRE}_{\text{LIDAR}}$ ($\sim 60$–$65 \, \text{W m}^{-2}$) occurs over the Tibet Autonomous Region (29° N, 97° E), where the opaque cloud cover is high (58 %) and the mean surface elevation is high ($\sim 4.42 \, \text{km}$) over 2.5 million $\text{km}^2$. Here, the high amount of moisture is uplifted towards southern Tibet, amplified by Rayleigh distillation as the vapor moves over the Himalayan mountains (He et al., 2015), which enhances the formation of opaque clouds.

CALIPSO space-based lidar differentiates well opaque clouds from thin clouds. Therefore, we can decompose the $\text{CRE}_{\text{LIDAR}}$ into contributions due to opaque clouds ($\text{CRE}_{\text{Opaque}}$: Fig. 8b) and thin clouds ($\text{CRE}_{\text{Thin}}$: Fig. 8c). This decomposition shows that 85 % ($23.0 \, \text{W m}^{-2}$) of the overall annual global mean $\text{CRE}_{\text{LIDAR}}$ ($27.0 \, \text{W m}^{-2}$) is produced by opaque clouds. Their effect is maximal ($\sim 50$–$55 \, \text{W m}^{-2}$) over the extra-tropical oceans (60° S and 60° N), where low

warm opaque clouds are numerous. Thin clouds contribute only 15 % ($4.0 \, \text{W m}^{-2}$) to the global $\text{CRE}_{\text{LIDAR}}$, and their effect is maximal ($\sim 13 \, \text{W m}^{-2}$) over the dry continental polar regions of the Greenland and Antarctic ice sheets, where the thin cloud cover is large ($\sim 40 \%$).

## 6 Evaluation of the new surface LW cloud radiative effect against ground-based stations

### 6.1 Method

Comparisons between ground-based measurements and the satellite-derived products ($\text{CRE}_{\text{LIDAR}}$, $\text{CRE}_{\text{2BFLX}}$) provide a direct evaluation of the satellite retrievals but are limited by the difference in the spatial resolution of the satellite-derived product ($2° \times 2°$) and the ground station observations (a few meters). For the satellite retrievals, we extract the monthly $2° \times 2°$ grid box centered at each ground site. For the ground-based observations, we extract the hourly observation at CALIPSO satellite overpass time above each ground site (two observations per day) and average over each month. We consider all days of each month, even if CALIPSO has no sampling over the site, because there are only a few days where CALIPSO observations are not available (e.g., 18 d TS2 in 2008). Moreover, in this study, we are interested in an accurate representation of the surface LW CRE interannual variability, which might have significant impacts on climate-relevant processes, and not only in an accurate representation of the anomalies observed in, e.g., Rutan et al. (2015). That CALIPSO is missing some sampling over the ground-based site will likely not significantly affect the interannual variability (i.e., months of maxima/minima of the surface LW CRE). The locations of the three ground-based sites are reported on the maps (stars in Fig. 14).

### 6.2 Time series

Over the Summit station Greenland site, on average compared to ground-based observations, LWCRE–LIDAR underestimates the surface LW CRE by $8.5 \, \text{W m}^{-2}$, while 2BFLX underestimates it by $16.4 \, \text{W m}^{-2}$ (Fig. 9a). Averages over the 2008–2010 and 2011–2015 periods (Fig. 10) show that these biases calculated for a short period are similar to the longer periods. Over the 2008–2011 time period, $\text{CRE}_{\text{LIDAR}}$ is close to $\text{CRE}_{\text{2BFLX}}$, and both show consistent summer maxima and winter minima, with $\text{CRE}_{\text{2BFLX}}$ slightly smaller than $\text{CRE}_{\text{LIDAR}}$ ($0.8 \, \text{W m}^{-2}$). Over the 2011–2015 time period, $\text{CRE}_{\text{LIDAR}}$ and the ground station data show similar annual cycles, and $\text{CRE}_{\text{LIDAR}}$ remains smaller than the Greenland site ($13.0 \, \text{W m}^{-2}$). In winter, the bias in $\text{CRE}_{\text{LIDAR}}$ can go up to $\sim 15 \, \text{W m}^{-2}$ compared to the Greenland site and is partly due to CALIPSO–GOCCP missing thin cloud below 2 km above ground level in winter, as shown in Lacour et al. (2017). While this comparison suggests that LWCRE–LIDAR could be biased somewhat low compared

**CALIPSO-GOCCP v3.1.2 OPAQ mask**

**Figure 7.** Pieces of the CALIPSO orbit passing over Africa on 11 August 2010 at 23:02:38 LST. Opaque clouds (left column) and thin clouds (right column). Top line: vertical feature mask from the CALIPSO–GOCCP–OPAQ product (Guzman et al., 20017); the black areas below 4 km correspond to land. Bottom line: the surface LWCRE–LIDAR. TS1

to the ground station perspective over Greenland, it is also clear that this approach captures the annual variability with a correlation coefficient between the $CRE_{LIDAR}$ and ground base site of 0.69 and a RMSE of $15.9\,\mathrm{W\,m^{-2}}$. The retrieval using $Z_{FA}$ instead of $Z_{T_{Opaque}}$ seems to compare to the Greenland ground-based observations more favorably (correlation coefficient of 0.70 and RMSE of $15.0\,\mathrm{W\,m^{-2}}$) with a smaller bias ($-11.6\,\mathrm{W\,m^{-2}}$ vs. $-13.6\,\mathrm{W\,m^{-2}}$, Table 1).

Over the mid-latitude continental site (Fig. 9b) on average, LWCRE–LIDAR underestimates the surface LW CRE by $5.7\,\mathrm{W\,m^{-2}}$ compared to ground-based observations with a correlation coefficient of 0.73 and RMSE of $11.0\,\mathrm{W\,m^{-2}}$, while 2BFLX underestimates it by $9.4\,\mathrm{W\,m^{-2}}$ with a correlation coefficient of 0.67 and RMSE of $15.5\,\mathrm{W\,m^{-2}}$.

Over the tropical ocean site (Fig. 9c) on average, LWCRE–LIDAR underestimates the surface LW CRE by $2.3\,\mathrm{W\,m^{-2}}$ compared to ground-based observations, and 2BFLX underestimates it by $4.1\,\mathrm{W\,m^{-2}}$. This same behavior is found on the map of differences between $CRE_{LIDAR}$ and $CRE_{2BFLX}$ (Fig. 14a) along the tropical Pacific and tropical Atlantic oceans, where 2BFLX underestimates the surface LW CRE compared to LWCRE–LIDAR.

### 6.3 Seasonal cycle

Figure 10 presents the comparison of seasonal cycles between the satellite retrievals and the ground-based observations.

Over the Greenland site (Fig. 10a, d), LWCRE–LIDAR and 2BFLX find the same seasonal cycle of the surface LW CRE with maxima in July that correspond to the maximum opaque cloud cover, same as the ground-based seasonal cycle.

Over the mid-latitude continental site (Fig. 10b, e), the surface LW CRE seasonal cycles of LWCRE–LIDAR and 2BFLX are close to each other, and the two satellite-derived products show similar seasonal cycles to the ground station.

Over the tropical ocean site (Fig. 10c, f), the surface LW CRE seasonal cycle is relatively flat.

### 6.4 Diurnal cycle

The time sampling is limited for LWCRE–LIDAR and 2BFLX as they observe each location only two times per day at about 01:30 and 13:30 local solar time (LST), and they do not implement diurnal variation correction in their algorithm. Nevertheless, diurnal variations of the cloud fraction profiles documented by the CATS/ISS lidar (Noel et al., 2018; Chepfer et al., 2019) indicate that the average of the cloud profiles

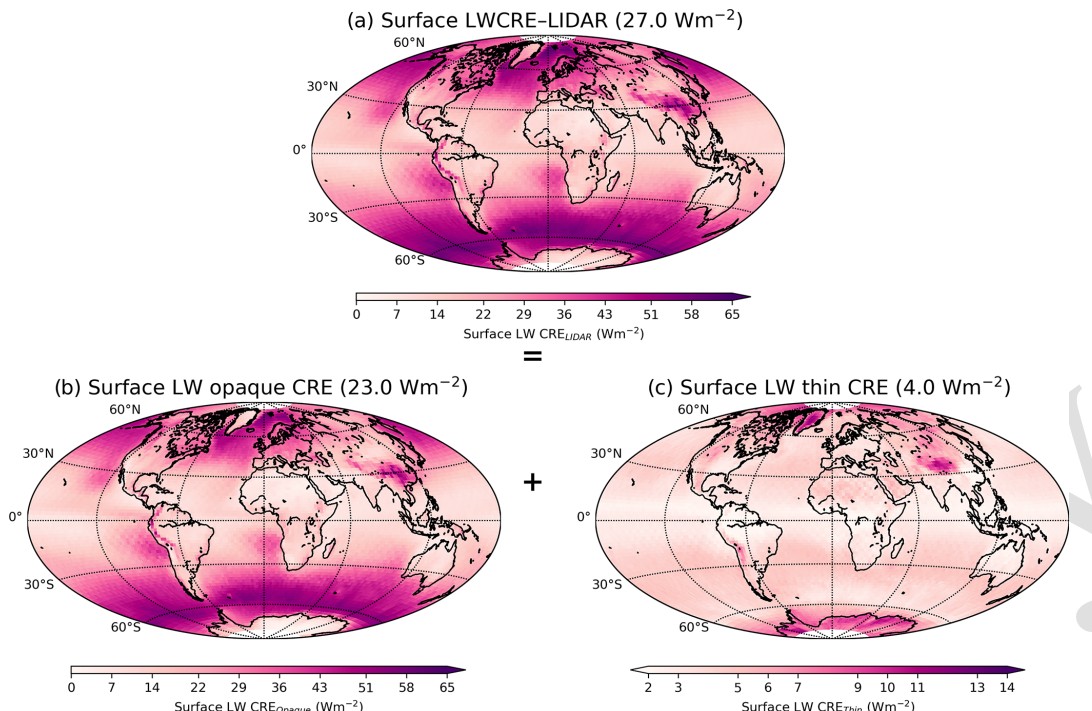

**Figure 8.** Maps of the surface LW CRE: **(a)** all clouds, **(b)** opaque clouds, and **(c)** thin clouds. This surface LW CRE is built from the CALIPSO–GOCCP v3.1.2 dataset (Fig. 2) and radiative transfer computations (Figs. 4–7, A1). The surface LW CRE is averaged over 2008–2020. Note that the color scale is different in panel **(c)**.

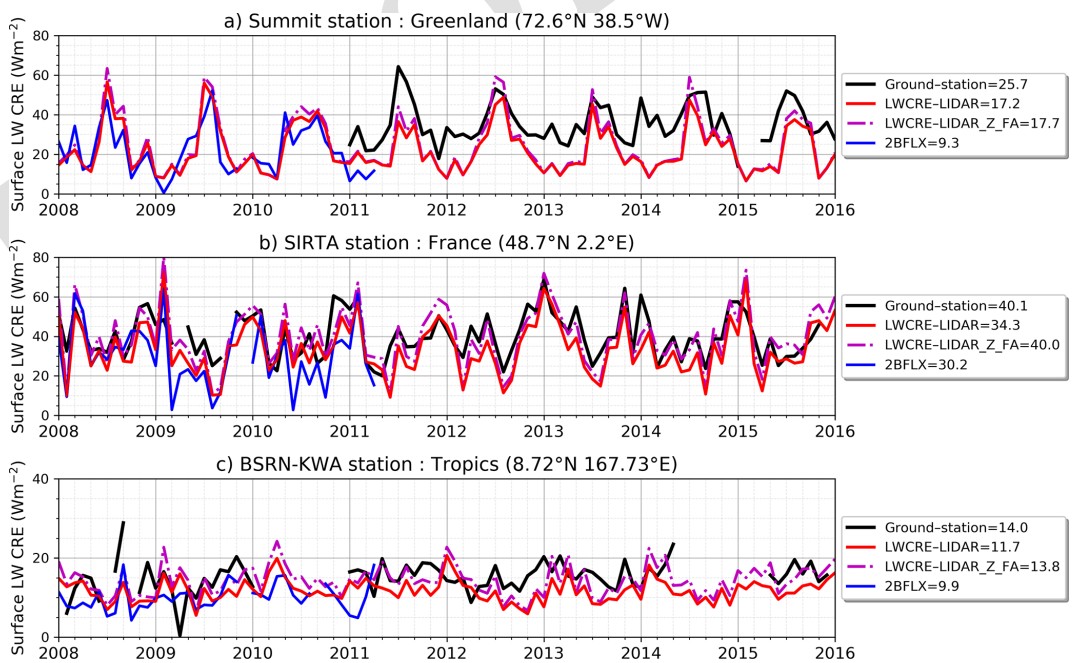

**Figure 9.** Comparisons between the surface LW CRE derived from ground station measurements and from satellites in three locations: **(a)** polar region at the Greenland Summit site, **(b)** mid-latitudes at the SIRTA site, and **(c)** tropics at the KWA site. Mean values reported in the legend are computed only over the time period when all products are available, e.g., only four months (Jan–Feb–Mar–Apr, 2011) for Greenland Summit mean values. The locations of the three sites are reported in Fig. 14. Note that the $y$-axis scale is different in each subplot.

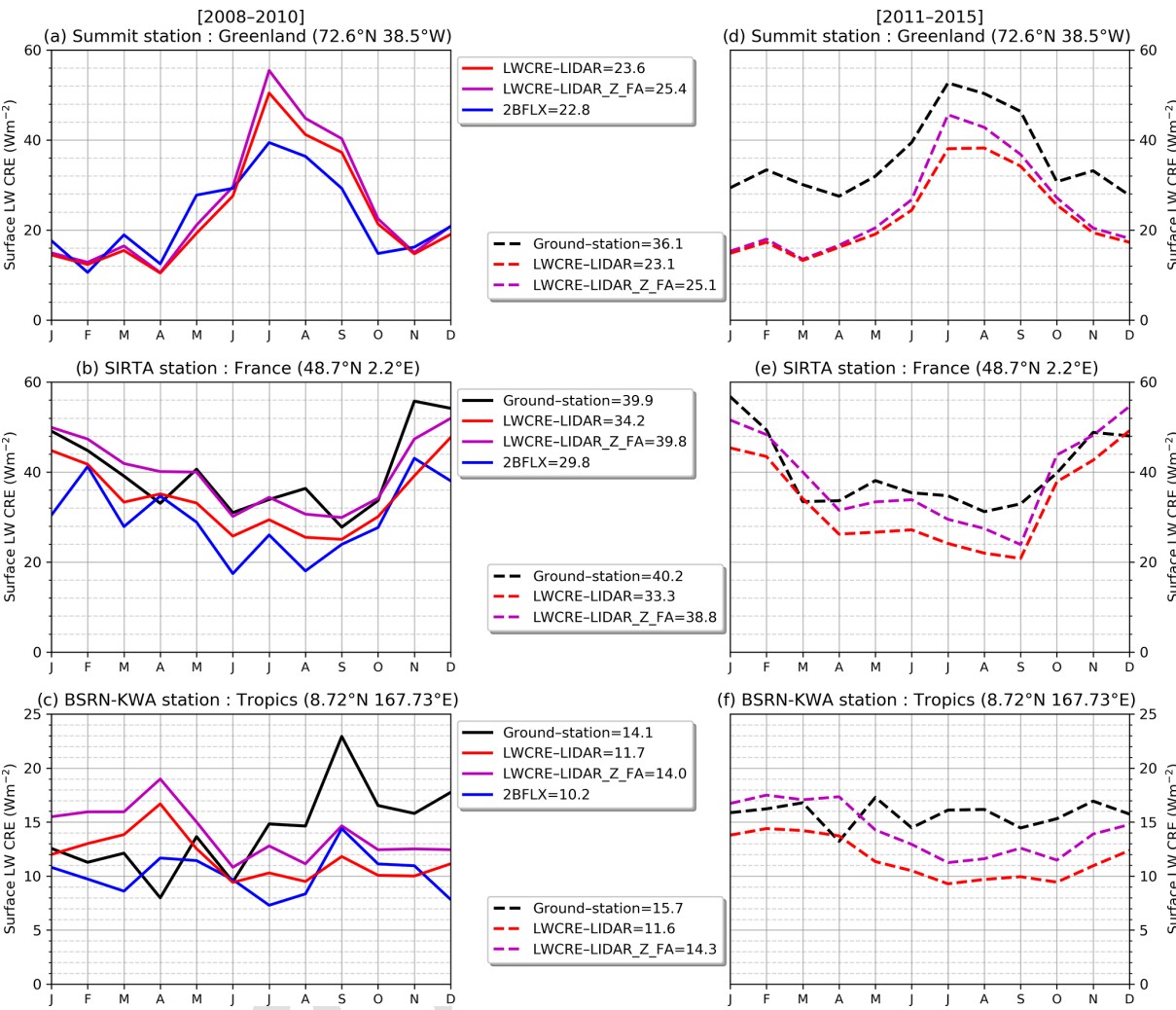

**Figure 10.** Same as Fig. 9 but in mean seasonal cycles. Panels **(a)**, **(b)**, and **(c)** correspond to 2008–2010 and panels **(d)**, **(e)**, and **(f)** correspond to 2011–2015. Note that the *y*-axis scale is different in each subplot.

**Table 1.** Bias, root-mean-squared error (RMSE) and correlation coefficient between satellite products and ground-based observations.

| | | 2008/01-2011/04 periode | | | 2008-2015 periode | |
|---|---|---|---|---|---|---|
| | | LWCRE–LIDAR | LWCRE–LIDAR_Z_FA | 2BFLX | LWCRE–LIDAR | LWCRE–LIDAR_Z_FA |
| Greenland | Bias | -8.5 | -7.9 | -16.4 | -13.6 | -11.6 |
| | RMSE | 9.0 | 8.4 | 16.9 | 15.9 | 15.0 |
| | Correlation | 0.91 | 0.95 | 0.45 | 0.69 | 0.70 |
| SIRTA | Bias | -5.7 | -0.1 | -9.9 | -6.6 | -0.8 |
| | RMSE | 11.0 | 10.4 | 15.5 | 10.8 | 9.5 |
| | Correlation | 0.73 | 0.73 | 0.67 | 0.77 | 0.77 |
| KWA | Bias | -2.3 | -0.3 | -4.1 | -3.4 | -0.9 |
| | RMSE | 6.1 | 5.6 | 6.9 | 5.7 | 4.9 |
| | Correlation | 0.03 | 0.15 | 0.23 | 0.08 | 0.21 |

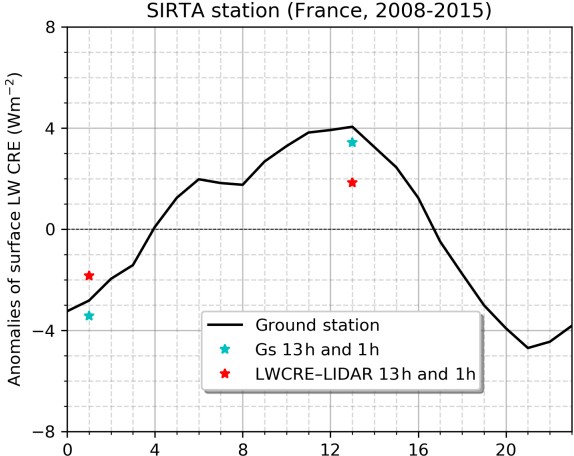

**Figure 11.** Same as Fig. 9b but in the anomaly of diurnal cycles over 2008–2015.

collected at 01:30 and 13:30 LST TS3 is similar to the average of all the profiles collected along the 24 h, with 13:30 LST corresponding to the minimum in cloud profiles along the day and 01:30 LST corresponding to the maximum (Fig. 7 in Noel et al., 2018). However, this statement is valid only between 55° S and 55° N.

Figure 11 shows the diurnal surface LW CRE variation observed at SIRTA in France, together with LWCRE–LIDAR. This comparison suggests that the average of the two CALIPSO overpasses each day is similar to the ground-based observed daily mean over this site. The absence of diurnal cycle correction might not be an important source of error in the LWCRE–LIDAR product.

# 7   Evaluation of the new surface LW cloud radiative effect against other satellite products

## 7.1   Comparison along pieces of orbits at footprint scale

CloudSat, CALIPSO, and CERES/Aqua satellites are part of the A-Train constellation (Stephens et al., 2002) and closely matched in time ($< 5$ min) and hence collocated by default, so we can compare them directly, assuming that the atmospheric changes occurring within 5 min are negligible.

Figure 12 shows a comparison between the surface LW CRE from the three spatial satellite retrievals along four pieces of orbits located over regions with different atmospheres and different surfaces. Figure 12 (top panel) shows the vertical CALIPSO–GOCCP cloud mask (Guzman et al., 2017), while Fig. 12 (bottom panel) represents the comparison between the surface LW CREs.

Orbit A passes over the eastern Pacific Ocean and observes a deep convective tower, a mid-level opaque cloud at an altitude of 7 km, and a low opaque cloud. The differences in surface LW CRE between the three spatial restitutions do not

exceed $\sim 5\,\mathrm{W\,m^{-2}}$. Nevertheless, within a small part of the orbit between 11.3 and 11.8° N (Fig. B1), we observe that the LWCRE–LIDAR retrieval is lower than the other two products, because the lidar does not detect a low cloud below $Z_{\mathrm{FA}}$, which is detected by CloudSat (shown in Fig. B1b).

Orbit B passes over the western Pacific Ocean and observes variable yet shallow clouds in the boundary layer ($< 2$ km). The $\mathrm{CRE_{LIDAR}}$ is intermittently larger than the other two products by $\sim 15\,\mathrm{W\,m^{-2}}$. CALIPSO–GOCCP (90 m cross track, 330 m along orbit track) detects shallow clouds in the boundary layer and, during the thickest of these, retrieves a surface LW CRE that is larger than the CERES (CloudSat) retrieval, which is based on a 20 km (5 km) footprint and might miss these clouds. TS5

Orbit C passes over ocean stratocumulus regions and observes a low opaque cloud. Between 12 and 19° S, the $\mathrm{CRE_{LIDAR}}$ ($\sim 60\,\mathrm{W\,m^{-2}}$) is smaller than $\mathrm{CRE_{2BFLX}}$ by $\sim 5\,\mathrm{W\,m^{-2}}$ and smaller than $\mathrm{CRE_{CERES}}$ by 15 W m$^{-2}$.

Orbit D passes over Antarctica and observes opaque clouds at high (10 km) and mid-level (4–5 km) altitudes. In the presence of high opaque clouds (between 68 and 71° S or between 73 and 77° S), $\mathrm{CRE_{LIDAR}}$ is lower than $\mathrm{CRE_{CERES}}$ by up to $\sim 20\,\mathrm{W\,m^{-2}}$ and $\mathrm{CRE_{2BFLX}}$ by up to $\sim 40\,\mathrm{W\,m^{-2}}$ but typically compares most favorably to $\mathrm{CRE_{CERES}}$ over the full scene.

These orbits show that by not including CloudSat, surface LWCRE–LIDAR is biased low by typically $\sim 10\,\mathrm{W\,m^{-2}}$ compared to 2BFLX and by $\sim 15\,\mathrm{W\,m^{-2}}$ compared to CERES-CCCM in regions of deep convection. In stratocumulus, surface LWCRE–LIDAR is biased low by typically $\sim 5\,\mathrm{W\,m^{-2}}$ compared to 2BFLX and $\sim 15\,\mathrm{W\,m^{-2}}$ compared to CERES-CCCM.

## 7.2   Global statistic at footprint scale over the ocean

Figure 13a shows a comparison between the surface LW CRE derived from the CALIPSO–GOCCP product (90 m cross track, 330 m along orbit track) collocated with CERES–CCCM that uses full-resolution CALIPSO and CloudSat data and reports the results over 20 km CERES footprints. We consider only the CERES–CCCM footprints where all the CALIPSO–GOCCP profiles falling within this footprint are opaque and where there are more than 40 profiles. To retrieve the surface LWCRE–LIDAR at the CERES–CCCM footprint resolutions, we average all $Z_{T_{\mathrm{Opaque}}}$ falling within CERES–CCCM's footprint and compute the surface LWCRE–LIDAR using the relationships found in Sect. 4.

We see a strong correlation between $\mathrm{CRE_{CERES}}$ and $\mathrm{CRE_{LIDAR}}$ ($R = 0.85$ TS6). Two significant departures from the one-to-one comparison line are observed: one for high values of the surface LW CRE and the second for low values. In the first pattern, for surface LW CRE greater than $\sim 70\,\mathrm{W\,m^{-2}}$, $\mathrm{CRE_{LIDAR}}$ is larger than $\mathrm{CRE_{CERES}}$. This pattern corresponds to some low marine opaque clouds in mid-latitude regions (not shown). To reconcile the two products,

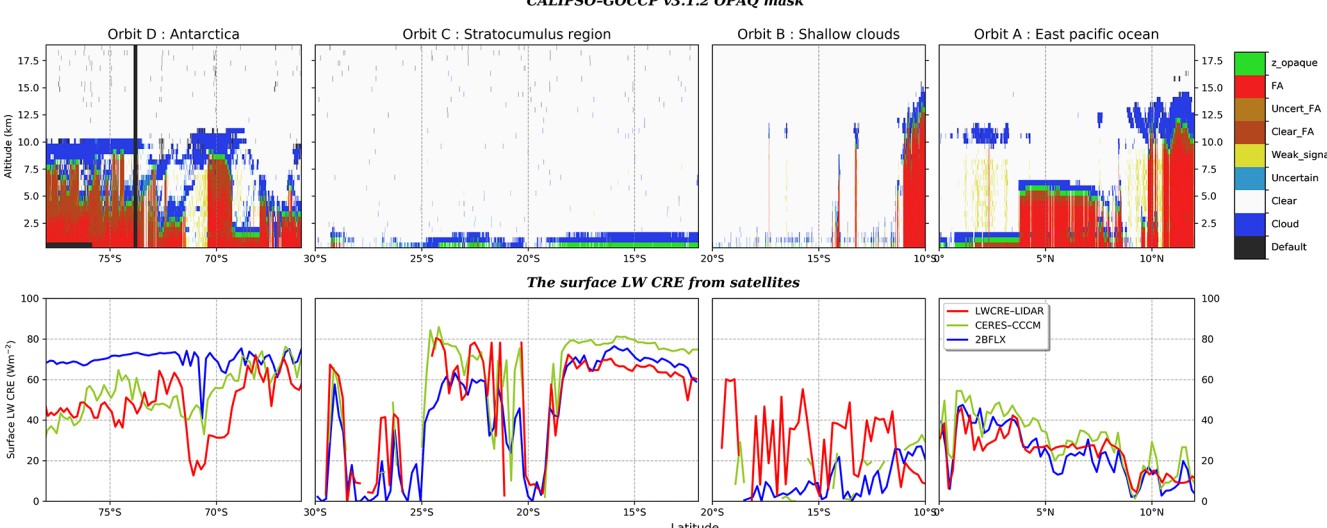

**Figure 12.** Pieces of CALIPSO orbits passing over **(a)** the eastern Pacific Ocean on 17 October at 08:21:48 LST, **(b)** the shallow cloud region in the Pacific Ocean on 5 April at 12:55:34 LST, **(c)** the stratocumulus region on 13 July at 06:48:37 LST, and **(d)** Antarctica on 21 September at 03:09:46 LST for the whole year 2008 TS4 . Top line: vertical feature mask from the CALIPSO–GOCCP–OPAQ product (Guzman et al., 20017); the black areas below 4 km correspond to land. Bottom line: surface LW CRE of the three satellite products. The locations of the pieces of orbit **(a, b, c, d)** are reported in Fig. 14.

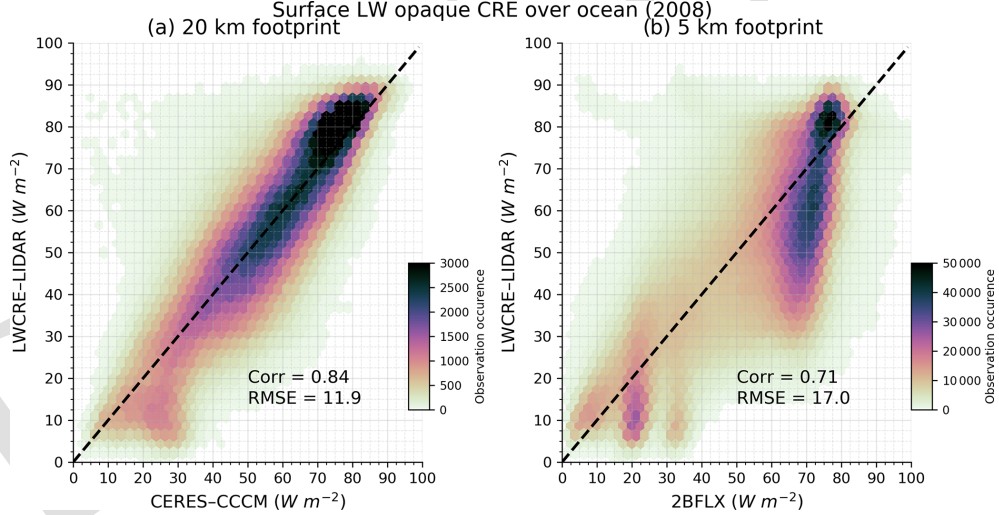

**Figure 13.** Instantaneous collocated surface LW opaque CRE at footprint scale: **(a)** LWCRE–LIDAR as a function of CERES–CCCM; **(b)** LWCRE–LIDAR as a function of 2BFLX. We only consider CERES (CloudSat) footprints where all CALIPSO footprints falling within the CERES (CloudSat) footprints are opaque and which contain at least 40 (10) profiles. Based on collocated observations over the ocean in 2008.

$CRE_{LIDAR}$ should be smaller by almost $\sim 5\,W\,m^{-2}$. One way to reduce this difference would be to increase the altitudes of clouds but, due to attenuation of the signal in opaque clouds, the space-based lidar would already potentially overestimate the overall height of the clouds. Thus, the cloud height is likely not the source of this difference. Another way to reduce the surface LW CRE is by decreasing the cloud cover or the cloud opacity. However, thanks to its high spatial resolution, the space-based lidar measures the cloud cover with

precision, and it should not overestimate the cloud opacity. Thus, the source of this apparent bias is more likely an underestimation of the humidity profiles used to retrieve the surface LW CRE in the presence of clouds. An increase in the humidity at these times would increase the $F^{net}_{Cloudy-freesky}$ and therefore decrease the surface LW CRE. A final possibility for the difference is that each product has a unique estimate of the cloud cover due to vastly different fields of view. CALIOP footprints are only a small fraction of the CERES

footprint, so part of the CERES footprint could be cloud-free even if the 40 CALIOP profiles are opaque. A study by Kato et al. (2010) demonstrated that the differences between CERES and CloudSat/CALIPSO cloud fractions decrease when averaged over area and time. Hence, this difference is likely not the primary source of bias when comparing large statistical datasets.

The second regime of differences among the products is for surface LW CRE less than $\sim 30\,\mathrm{W\,m^{-2}}$ (Fig. 13a), which corresponds to high opaque clouds over the warm pool region (not shown). Here, $\mathrm{CRE_{LIDAR}}$ is smaller than $\mathrm{CRE_{CERES}}$. The underestimation of surface LWCRE–LIDAR compared to CERES–CCCM could be caused by the full attenuation of the laser beam in deep convective clouds such that CALIPSO–GOCCP overestimates the mean altitude of opaque clouds.

Figure 13b represents the comparison between the surface LW CRE derived from the CALIPSO–GOCCP product (90 m cross track, 330 m along orbit track) collocated with the CloudSat 2BFLX product at a resolution of the CloudSat footprint (5 km). We also consider only the CloudSat footprints where all the CALIPSO–GOCCP profiles falling within this footprint are opaque and where there are more than 10 profiles, and we compute $\mathrm{CRE_{LIDAR}}$ by averaging all $Z_{T_{\mathrm{Opaque}}}$ falling within the CloudSat footprint.

Three significant departures from the one-to-one comparison line are observed: one for low values where $\mathrm{CRE_{LIDAR}} < \mathrm{CRE_{2BFLX}}$, one for high values of the surface LW CRE where $\mathrm{CRE_{LIDAR}} > \mathrm{CRE_{2BFLX}}$, and one for high values where $\mathrm{CRE_{LIDAR}} < \mathrm{CRE_{2BFLX}}$. The first two patterns appear to be similar to Fig. 13a and show up for the same reasons as described above. The last pattern of differences among the products is for large values of surface LW CRE where $\mathrm{CRE_{2BFLX}}$ is larger than $\mathrm{CRE_{LIDAR}}$. This pattern corresponds to a subsample of marine opaque clouds (25 % of the opaque cloud collocated) in mid-latitude regions (not shown) where CloudSat is able to detect lower clouds than CALIPSO. Using $Z_{\mathrm{FA}}$ instead of $Z_{T_{\mathrm{Opaque}}}$ in LWCRE–LIDAR retrieval would shift this pattern upward and reduce the sample (17 % vs. 25 %).

The differences shown in Fig. 13 are expected when comparing satellite products at footprint scales that use different remote sensing techniques. However, when looking at the gridded product distributions (Fig. B3) instead of instantaneous collocated data, the 2BFLX and LWCRE–LIDAR agree well.

## 7.3 Global mean comparison at gridded scale

To compare 2BFLX and LWCRE–LIDAR at gridded scale, we averaged 2BFLX initially at $1° \times 1°$ resolution to $2° \times 2°$ like the CALIPSO–GOCCP product.

Figure 14a shows global maps of differences between LWCRE–LIDAR and 2BFLX. This comparison gives an overview of the differences between the two surface LW

CRE spatial products, but it may mask some differences given the fact that the two spatial products are averaged in time (monthly) and space ($2° \times 2°$ latitude–longitude gridded).

In the global annual mean, $\mathrm{CRE_{LIDAR}}$ is slightly higher compared to $\mathrm{CRE_{2BFLX}}$ ($0.7\,\mathrm{W\,m^{-2}}$).

Compared to 2BFLX (Fig. 14a), $\mathrm{CRE_{LIDAR}}$ is slightly larger than $\mathrm{CRE_{2BFLX}}$ over tropical oceans. Over lands, $\mathrm{CRE_{LIDAR}}$ is slightly lower than $\mathrm{CRE_{2BFLX}}$. The maximum difference occurs over land along the western coasts of the North and South American continents and the Himalayan mountains, where the surface elevation is above 2.5 km. This difference might be due to the CloudSat CPR's long powerful pulse (Fig. B2), which generates a surface clutter echo that tends to partially mask signals from clouds forming below $\sim 1$ km (Marchand et al., 2008). Over icy polar areas, the two products are very similar.

Zonal averages of the surface LW CRE for 2008–2010 (Fig. 14c) show that the surface LW CRE is generally low in tropical regions and increases towards the mid-latitudes as the atmospheric moisture decreases. Values do not vary much northward of about $50°$ N. To the south, a maximum occurs at about $60°$ S, with a decline towards the far south due to less cloudiness. Over the broad domain, reaching from $60°$ N to $60°$ S, the two satellite techniques show similar zonal means, with differences among the two typically not exceeding $\sim 3\,\mathrm{W\,m^{-2}}$.

## 7.4 Variations of 13 years (2008–2020)

Figure 15a shows the temporal evolution of the surface LW CRE anomaly from the two satellite-derived products. A decomposition, separating continents from oceans and Northern Hemisphere (NH) from Southern Hemisphere (SH), is presented in Fig. 17b–g.

The phasing of the annual cycle of $\mathrm{CRE_{LIDAR}}$ and $\mathrm{CRE_{2BFLX}}$ anomalies is roughly similar over the 2008–2010 time period. The phasing of the annual cycle of the two products is actually quite consistent for both the NH and SH over both land and ocean. For the NH land (Fig. 17d), the $\mathrm{CRE_{2BFLX}}$ is slightly larger than $\mathrm{CRE_{LIDAR}}$. However, it is interesting that even over NH land the annual minima match pretty well.

The interannual variability is pretty interesting. For example, the NH winter maximum in LWCRE–LIDAR products appears to vary by up to about $\sim 3\,\mathrm{W\,m^{-2}}$ from year to year. That is the kind of variability that might have significant impacts on climate-relevant processes like melting of the cryosphere.

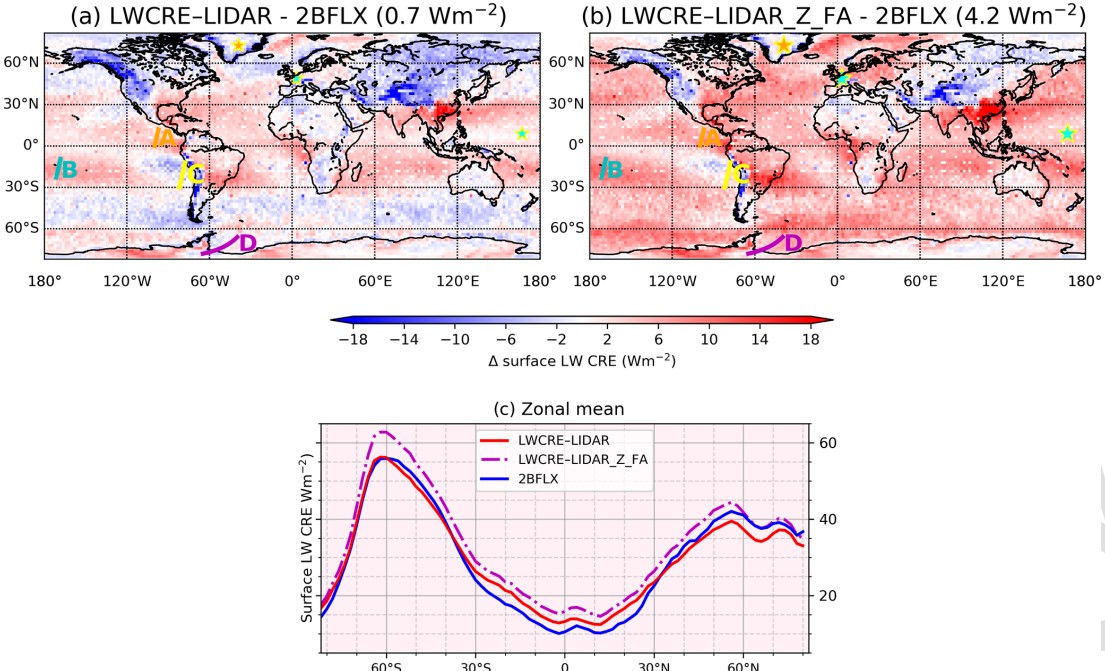

**Figure 14.** Maps of differences in the surface LW CRE **(a)** LWCRE–LIDAR minus 2BFLX, **(b)** LWCRE–LIDAR_Z_FA minus 2BFLX, and **(c)** zonal means of the two satellite products. Data are averaged over 2008–2010. Locations of the three ground-based sites and pieces of orbits are reported on the maps.

## 8 Discussion

### 8.1 About the space lidar missing the opaque cloud base

Based on the comparison of orbits (Fig. B1), we observe that when the space lidar does not see the cloud base, LWCRE–LIDAR underestimates the local surface LW CRE compared to 2BFLX. However, the deep convective opaque clouds cover a small part of the overall tropics compared to other clouds. Therefore, this effect does not dominate the global comparison (Fig. 14a), where surface LWCRE–LIDAR is contrarily slightly larger than the other satellite product. Figure 13b (where 2BFLX is about $\sim 70\,\mathrm{W\,m^{-2}}$) and Fig. 14a (Southern Ocean) consistently suggest that CALIPSO not seeing the cloud base leads to LWCRE–LIDAR underestimating the surface LW CRE more frequently in the extratropical oceanic storm tracks than elsewhere.

To test whether the differences between LWCRE–LIDAR and other satellite products come from the space lidar not seeing the cloud base, we used two different approaches.

First, we used $Z_{\mathrm{FA}}$ instead of $Z_{T_{\mathrm{Opaque}}}$ in the LWCRE–LIDAR retrieval. By definition, $Z_{\mathrm{FA}}$ is always lower in altitude than $Z_{T_{\mathrm{Opaque}}}$. Therefore, this change should reduce the difference between the surface LWCRE–LIDAR and other surface LW CREs if the differences were due to CALIPSO missing the cloud base. Figure 14b shows that the difference between surface LWCRE–LIDAR and the other satel-

lite product increases instead of decreases when using $Z_{\mathrm{FA}}$ instead of $Z_{T_{\mathrm{Opaque}}}$. This suggests that the differences in surface LW CRE are likely not often due to CALIPSO misrepresenting the cloud base and that, in the majority of the cases, the cloud base might not be far from $Z_{\mathrm{FA}}$. Nevertheless, contrary to the satellite retrieval intercomparison, using $Z_{\mathrm{FA}}$ instead of $Z_{T_{\mathrm{Opaque}}}$ leads to slightly better agreement between LWCRE–LIDAR and ground-based retrievals (e.g., Figs. 9, 10, Table 1). Ground-based measurements derive directly the surface LW CRE. While there are certainly challenges in comparing ground-based and satellite estimates, we should consider the ground-based estimates to be of pretty high quality.

Second, we used the cloud-base height (called the CBASE dataset) described in Mülmenstädt et al. (2018) instead of $Z_{\mathrm{FA}}$ to compute $Z_{T_{\mathrm{Opaque}}}$. In the CBASE dataset, the cloud-base-height value is given at a horizontal resolution of 40 km along the CALIPSO orbit track in the portion of the orbit where clouds are opaque. Along each CALIPSO orbit, we collocated the cloud-base-height dataset with the CALIPSO–GOCCP dataset and replaced $Z_{\mathrm{FA}}$ with the cloud-base-height value given in the CBASE dataset. Then we computed $Z_{T_{\mathrm{Opaque}}}$ and the surface LW CRE. CBASE values are distributed at all latitudes and are available in 33.2 % of all the CALIPSO opaque profiles because CBASE can be retrieved only when thin clouds are detected within the 40 km orbit piece that also contains opaque cloud profiles. Comparing Fig. 16a and b indicates that the subsample of opaque

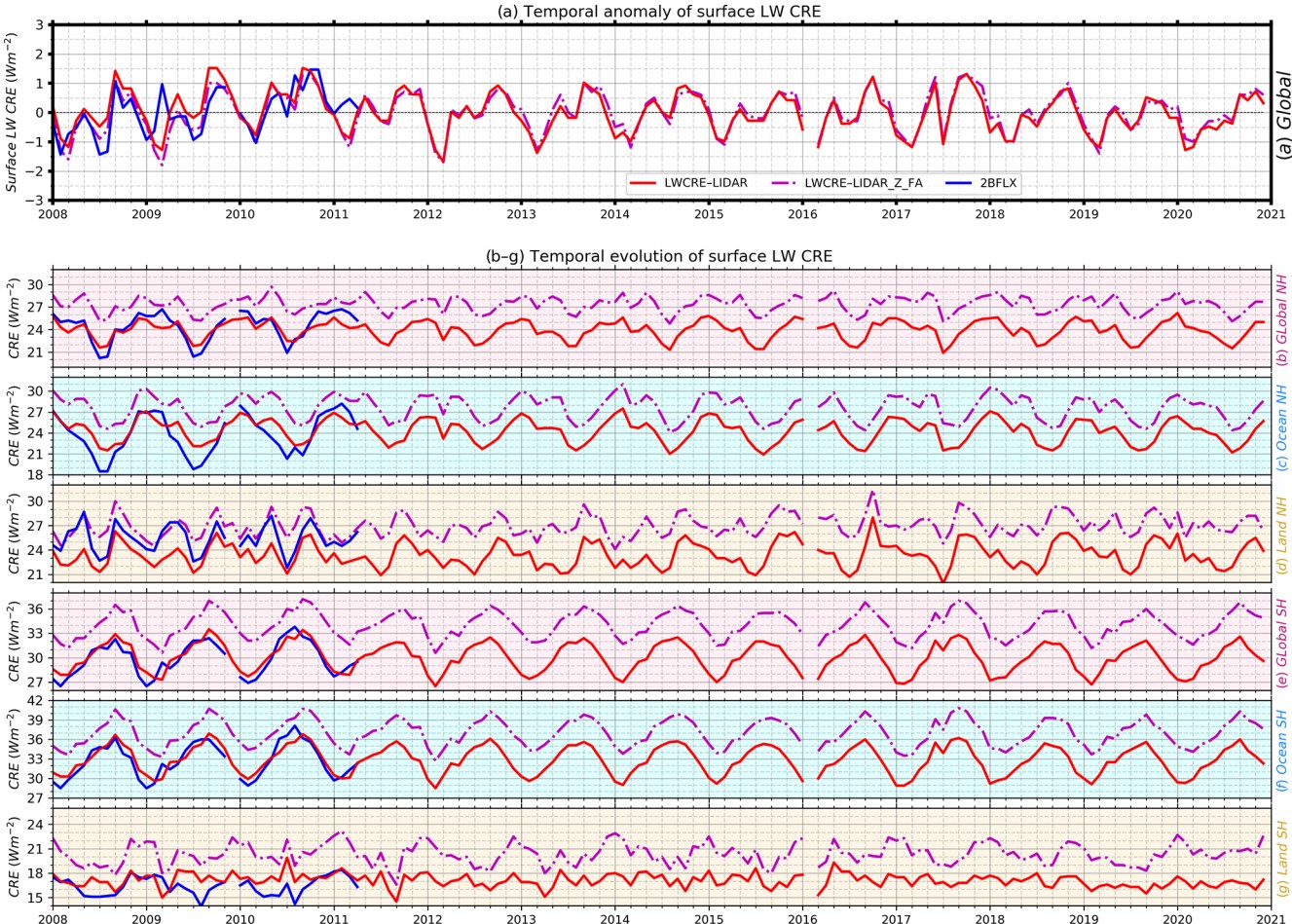

**Figure 15. (a)** Time series of global surface LW CRE anomalies. **(b–f)** Time series of surface LW CREs over all NH, ocean NH, land NH, all SH, ocean SH, and land SH. In panel **(a)** the anomaly is defined as the global average for each month of each product minus its own average over the whole time series. Note that the $y$-axis scale is different in each subplot.

CALIPSO–GOCCP profiles where CBASE is documented contains both large values of surface LW CRE associated with mid- and low-level clouds located at mid-latitudes (upper right data in panel b) and small values of surface LW CRE (lower left), but it does not include the data where 2BFLX is much larger than LWCRE–LIDAR which correspond to mid-latitude oceanic opaque clouds. When replacing $Z_{FA}$ (Fig. 16b) with CBASE (Fig. 16c) in the LWCRE–LIDAR algorithm, the surface LWCRE–LIDAR rises slightly almost everywhere because CBASE is lower in altitude than $Z_{FA}$, and surface LWCRE–LIDAR values lower than $\sim 18\,\mathrm{W\,m^{-2}}$ are no longer present. The latter correspond to both deep convective clouds and shallow boundary layer clouds. The correlation between 2BFLX and LWCRE–LIDAR is similar whether we use $Z_{FA}$ (0.79) or CBASE (0.78) in the LWCRE–LIDAR algorithm.

This sensitivity study suggests that using a more advanced cloud base height (here CBASE) derived from lidar measurements than $Z_{FA}$ in the LWCRE–LIDAR algorithm will in-

crease the surface LW CRE value retrieved in some opaque cloud profiles slightly, but it does not fundamentally change the results.

Thus, what these results mean collectively is that (1) the inability of CALIPSO to observe the cloud base likely does have some effect (with respect to ground-based measurements). (2) This effect actually makes the comparison to other satellite products worse, which means that there are other issues (possibly also with the other satellite product), leading to further differences.

## 8.2 About the sub-daily variability of the humidity and temperature profiles

Looking for other issues that could explain the differences between satellite products, we examined humidity and temperature profiles.

Contrary to the surface LWCRE–LIDAR retrieval method, CERES-CCCM and 2BFLX retrievals of surface flux account for sub-daily variations in temperature/humidity and

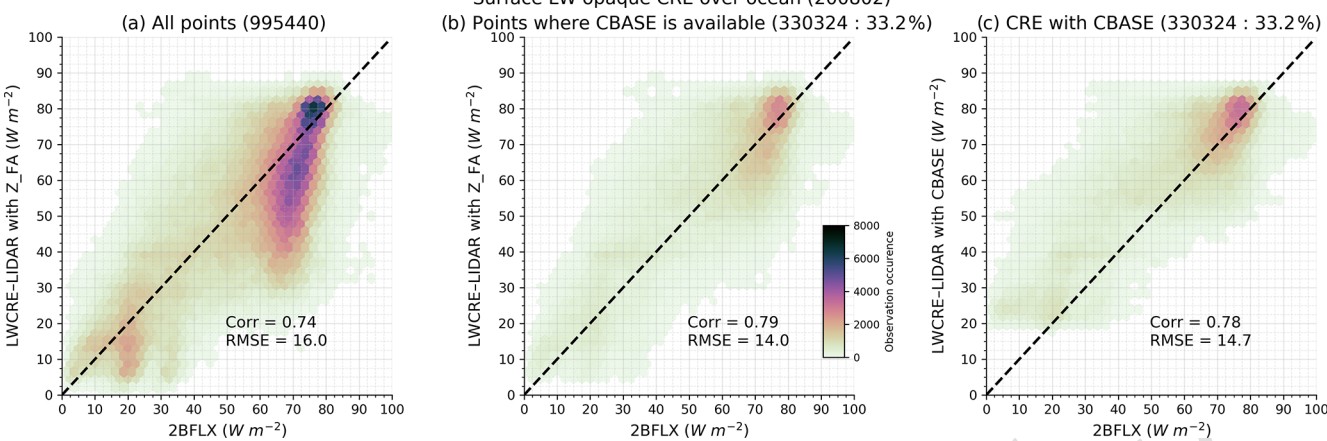

**Figure 16.** Surface LW CRE derived from LWCRE–LIDAR as a function of the one derived from 2BFLX. **(a)** Surface LWCRE–LIDAR (*y* axis) computed using the altitude of full lidar attenuation **(b)**, same as **(a)** but containing only the subsample of CALIPSO profiles where cloud-base-height values are available from Mülmenstädt et al. (2018). **(c)** Same as **(b)**, but the surface LWCRE–LIDAR is computed using the cloud-base-height values from Mülmenstädt et al. (2018) instead of the altitude of lidar full attenuation. The color scale indicates the number of occurrences at 5 km resolution (footprint scale of CloudSat) over ocean in February 2008.

capture regional variations (e.g., eastern vs. western tropical Pacific), climate events (e.g., ENSO), and extreme changes over polar regions. 2BFLX uses 3-hourly atmospheric state variable data on a half-degree Cartesian latitude and longitude grid from AN-ECMWF.

As shown in Fig. 14, monthly mean gridded surface LWCRE–LIDAR is consistent with 2BFLX, even though 2BFLX uses sub-daily spatiotemporal resolutions of temperature/humidity profiles, while LWCRE–LIDAR uses monthly mean temperature/humidity profiles. Nevertheless, instantaneous surface LWCRE–LIDAR retrievals are likely more biased than the monthly mean gridded surface LWCRE–LIDAR due to the use of monthly mean temperature/humidity profiles, because monthly means miss extreme humidity and temperature profiles. To estimate the error in the instantaneous surface LWCRE–LIDAR values, we compared the instantaneous surface LWCRE–LIDAR values obtained using 6-hourly temperature/humidity profiles from ERA-I to one obtained using monthly means and 2BFLX, for 1 d at footprint scales.

Figure 17 shows that using sub-daily profiles in LWCRE–LIDAR retrieval makes the comparison to other satellite products worse at footprint scale. More analysis (not shown) indicates that, for thin clouds, surface LWCRE–LIDAR retrieved using sub-daily temperature/humidity profiles agrees better with 2BFLX (at 5 km resolution) than surface LWCRE–LIDAR retrieved using monthly mean profiles. In contrast, the agreement between LWCRE–LIDAR and other products is lower when using sub-daily temperature/humidity profiles in all other cases: opaque clouds and also thin clouds when compared to CERES-CCCM (at 20 km). Overall, this suggests that the differences between the three daily products are likely due to other causes than

LWCRE–LIDAR using monthly mean temperature/humidity profiles.

## 9 Conclusions

In this paper, we build a new surface LWCRE–LIDAR dataset from five cloud properties observed with space-based lidar (CALIPSO–GOCCP product). The robustness of the new surface LWCRE–LIDAR dataset at global scales is evaluated by comparing it to existing independent space-based surface LW CRE retrievals from CERES and CloudSat (Kato et al., 2010; L'Ecuyer et al., 2019) at the instantaneous footprint scale as well as at the $2° \times 2°$ gridded global scale. It is also evaluated locally by comparison to observations collected at three ground stations in polar (Shupe et al., 2013), mid-latitude (Haeffelin et al., 2005; Chiriaco et al., 2018), and tropical (Roesch et al., 2011) locations. The (admittedly limited) ground station comparisons actually showed that the LWCRE–LIDAR product agreed best with the ground measurements compared to the other satellite product. It appears that it captures the interannual variability well. Additionally, there are other specific aspects where the LWCRE–LIDAR product appears to be an improvement over others in providing a longer time series, including over bright polar surfaces.

This might be surprising given the simplicity of the surface radiation retrieval method used to produce the LWCRE–LIDAR product, but this is understandable because of the following two physical elements.

i. The LWCRE–LIDAR method directly retrieves the surface LW CRE without retrieving the surface radiative fluxes first. This approach minimizes the impact of the uncertainties due to surface characteristics (sur-

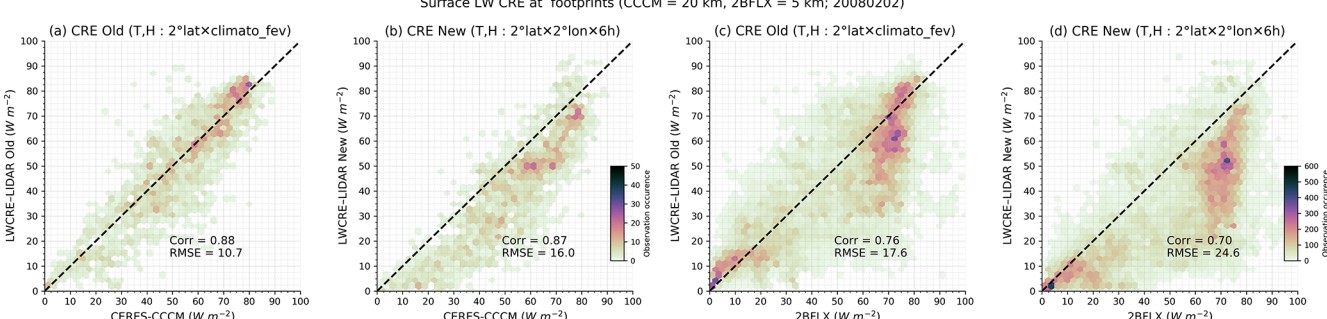

**Figure 17.** Distribution of the surface LW CRE: **(a)** LWCRE–LIDAR retrieved using monthly mean temperature/humidity profiles as a function of CERES-CCCM with data at 20 km resolution (CERES SSF footprint); **(b)** same as **(a)** but LWCRE–LIDAR retrieved using sub-daily temperature/humidity profiles; **(c, d)** same as **(a, b)** but for 2BFLX instead of CERES-CCCM and using data at 5 km resolution (CloudSat resolution).

face emissivity, roughness, deserts and frozen surfaces), which strongly influence the fluxes but not the surface LW CRE.

ii. The surface LW CRE is primarily driven by the cloud cover, the cloud opacity, and the cloud altitude, which are documented by space-based lidar over all types of surfaces. Moreover, the lidar approach distinguishes quite well the opaque clouds from the optically thin clouds. Lastly, it documents the detailed vertical cloud profile, except below the altitude where the laser is fully attenuated, where we overestimate the mean altitude of opaque clouds. This last limitation only weakly influences the surface LW CRE retrieval because the lidar is fully attenuated at an altitude lower than 3 km above the surface most of the time (Guzman et al., 2017), except in deep convection and some mid-latitude clouds, indeed, in deep convective tropical regions where the attenuation of the lidar beam might not see the whole bottom part of the cloud and can underestimate the surface LW CRE by almost $\sim 5\,\mathrm{W\,m^{-2}}$. All three satellite datasets exhibit some differences relative to ground-based measurements and can go up to $\sim 15\,\mathrm{W\,m^{-2}}$ bias in the surface LW CRE over polar regions. The $\sim 15\,\mathrm{W\,m^{-2}}$ bias in LWCRE–LIDAR over Summit in winter is partly due to CALIPSO–GOCCP missing thin cloud below 2 km above ground level in winter, as shown in Lacour et al. (2017).

The evaluation of this new surface LWCRE–LIDAR against other datasets also showed that (overall) this new retrieval agrees well with CloudSat-based estimates (L'Ecuyer et al., 2019) and CERES–CCCM, but the latter are limited in time until only 2011 due to a battery anomaly.

This new global dataset extends over more than a decade thanks to the long CALIPSO mission. The global mean temporal evolution over 13 years (2008–2020) shows that the maximum anomaly of the surface LWCRE–LIDAR in the NH winter varies by up to about $\sim 3\,\mathrm{W\,m^{-2}}$ from year to

year. This new dataset will be extended in time by including future data acquired by CALIPSO as well as data collected by forthcoming space lidars on board the European Earth Cloud, Aerosol and Radiation Explorer mission (Earth-CARE; Illingworth et al., 2015) and the next generation of US cloud/aerosol lidar space missions if we are able to reconcile data from successive space lidar missions. The monthly gridded dataset is available for the 2008–2020 time period at https://doi.org/10.14768/70d5f4b5-e740-4d4c-b1ec-f6459f7e5563 (Arouf et al., 2022).

The dataset presented in this paper will be used in a future study to better understand the mechanisms of cloud radiative feedbacks at the Earth's surface, i.e., how a change in surface temperature modifies the cloud properties that change the surface LW CRE, which in turn influences the temperature. An essential first step is to understand which cloud variables have driven the surface LW CRE variations over the last decade in regions that are most sensitive to global warming, such as the polar regions, as well as on a global scale. Several recent studies (e.g., Taylor et al., 2007; Zelinka et al., 2012a, b; Vaillant de Guélis et al., 2017a, b, 2018) have shown that it is possible to attribute changes in CRE to variations in cloud properties when (1) the CRE is related to a limited number of cloud properties by sufficiently simple relationships that they can be derived analytically, (2) the CRE retrieved by these analytical relationships is sufficiently reliable, i.e., within the uncertainty domain of the existing datasets, and (3) the CRE is retrieved using reliable observations over all surface types and on a long global timescale. The surface LWCRE–LIDAR dataset developed in this study satisfies these three conditions. The next step of this work will therefore be to analyze this 13-year dataset to understand these mechanisms. The goal of this research is to improve our understanding of the response of clouds to the warming induced by anthropogenic activities, which is a major source of uncertainty in climate change predictions.

## Appendix A: Sensitivity of the surface LW CRE to humidity and temperature

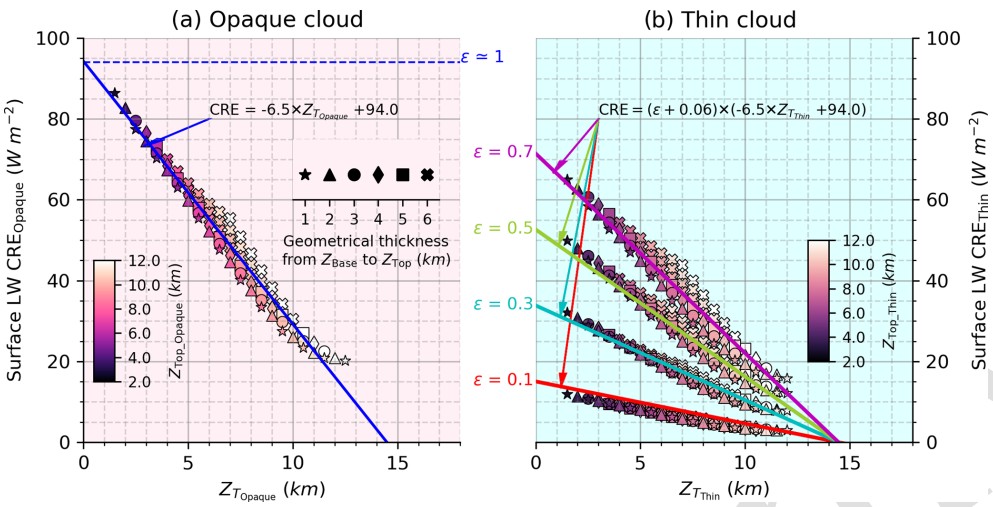

**Figure A1.** Same as Fig. 3 but over land.

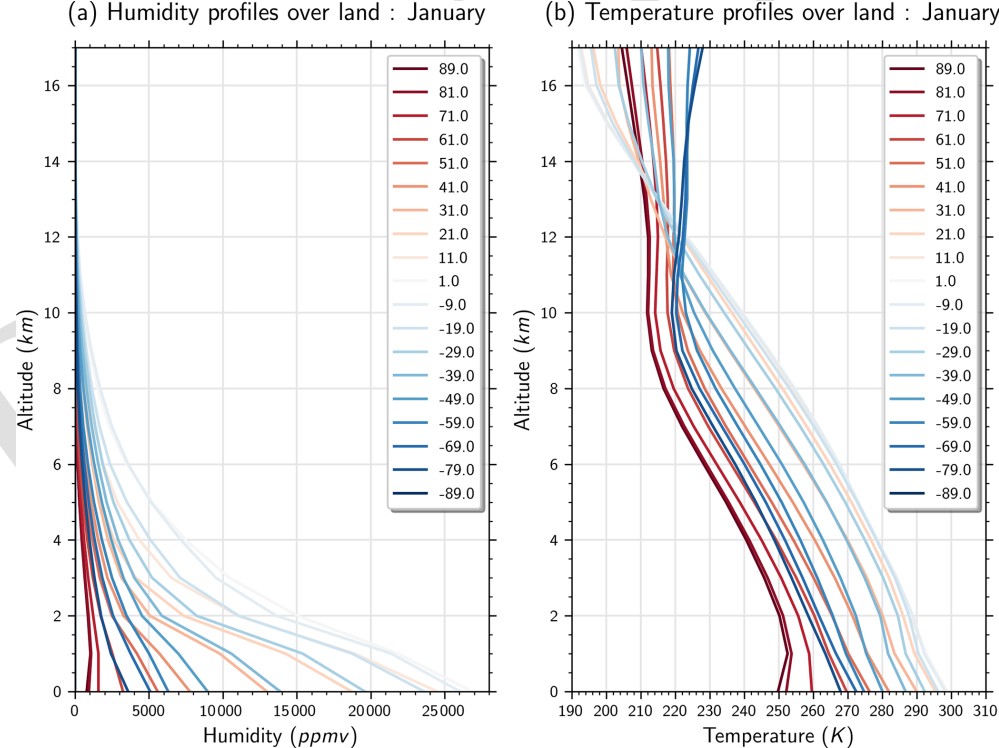

**Figure A2.** Example of ERA-Interim atmospheric profiles taken over land in January and averaged over $10°$ latitude bands.

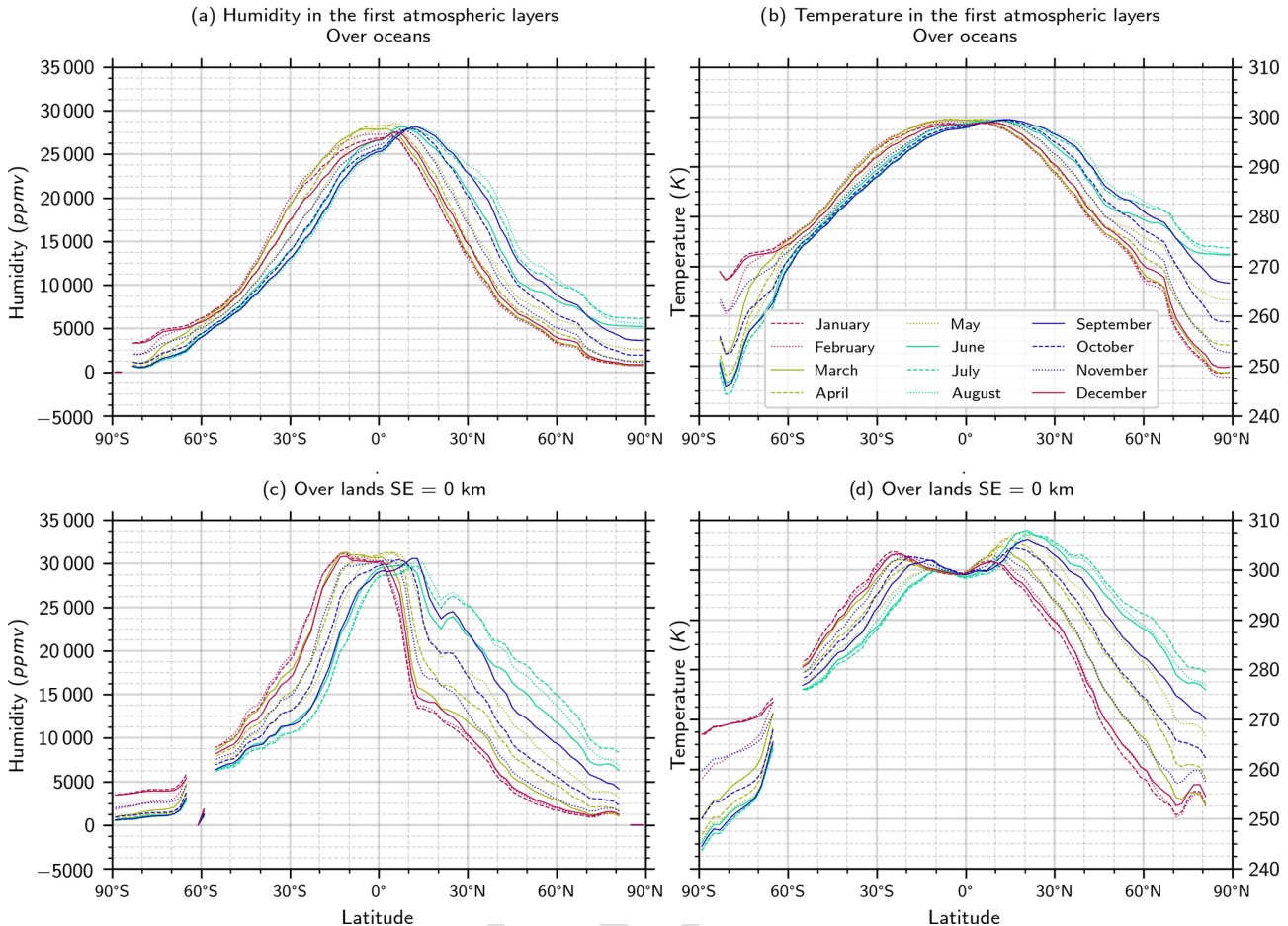

**Figure A3.** Seasonal and zonal variations of the temperature and humidity in the near-surface atmospheric layer ($Z < 1$ km) from ERA-Interim.

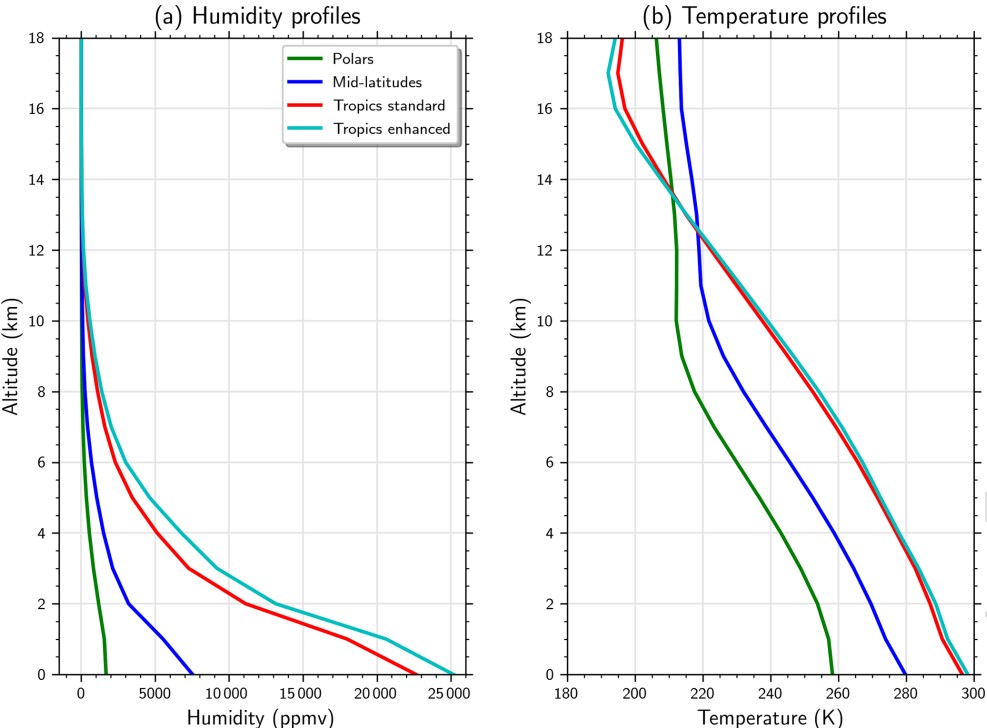

**Figure A4.** Annual mean profiles of temperature and humidity from ERA-Interim.

**Appendix B: Sensitivity of the surface LW CRE to cloud base height**

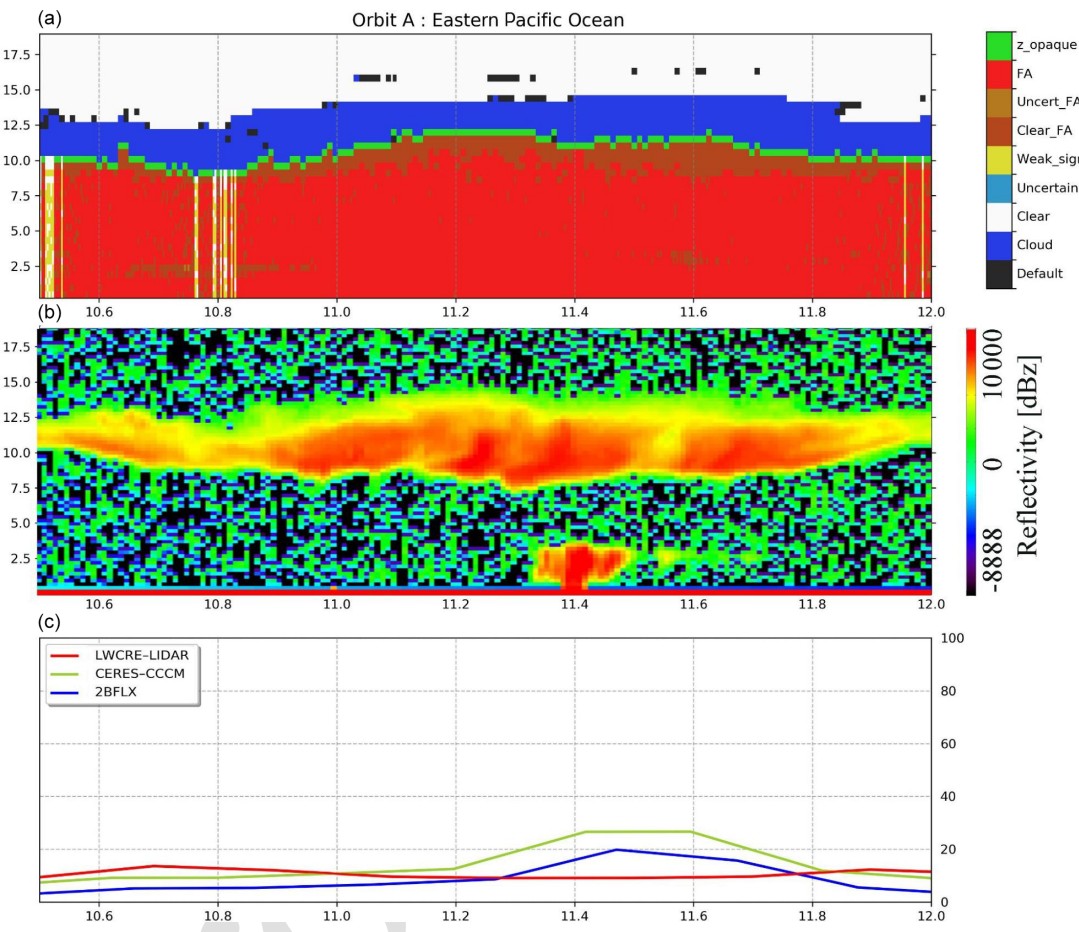

**Figure B1.** Same as Fig. 12 orbit A between 10.5 and 12° N: **(a)** CALIPSO–GOCCP–OPAQ mask, **(b)** CloudSat reflectivity, and **(c)** surface LW CREs.

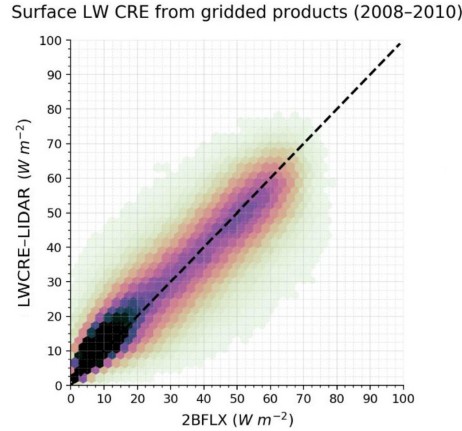

**Figure B2.** Same as Fig. B1 but for a piece of orbit passing over China on 10 November 2008 at 18:58:39 LST.

**Figure B3.** Comparison of monthly $2° \times 2°$ gridded surface LW CRE from LWCRE–LIDAR and 2BFLX.

## Appendix C

**Table C1.** LWCRE–LIDAR–Ed1 monthly gridded products: definitions and variable names.

| Geophysical quantity | Variable name in the nc file | Unit | Dim |
|---|---|---|---|
| Time | time | Month | time |
| Longitude | lon | $^\circ$ E | long |
| Latitude | lat | $^\circ$ N | lat |
| Surface cloud radiative effects net longwave flux monthly means (surface LW CRE) | sfc_cre_net_lw_mon | W m$^{-2}$ | (Time, lat, long) |
| Surface opaque cloud radiative effects net longwave flux monthly means (surface LW opaque CRE) | sfc_cre_net_lw_mon_opaque | W m$^{-2}$ | (Time, lat, long) |
| Surface thin cloud radiative effects net longwave flux monthly means (surface LW thin CRE) | sfc_cre_net_lw_mon_thin | W m$^{-2}$ | (Time, lat, long) |
| Surface cloud radiative effects net longwave flux monthly means derived using fully attenuated altitude in opaque scenes (surface LW CRE_Z_FA) | sfc_cre_net_lw_mon_Z_FA | W m$^{-2}$ | (Time, lat, long) |
| Top of the atmosphere cloud radiative effects longwave flux monthly means (TOA LW CRE) | toa_cre_lw_mon | W m$^{-2}$ | (Time, lat, long) |
| Top of the atmosphere opaque cloud radiative effects longwave flux monthly means (TOA LW opaque CRE) | toa_cre_lw_mon_opaque | W m$^{-2}$ | (Time, lat, long) |
| Top of the atmosphere thin cloud radiative effects longwave flux monthly means (TOA LW thin CRE) | toa_cre_lw_mon_thin | W m$^{-2}$ | (Time, lat, long) |
| CALIPSO opaque cloud cover (C_Opaque) | cltcalipso_opaque | % | (Time, lat, long) |
| CALIPSO opaque cloud altitude (Z_T_Opaque) | cltcalipso_opaque_z | km | (Time, lat, long) |
| CALIPSO fully attenuated altitude (Z_FA) | zopaque | km | (Time, lat, long) |
| CALIPSO thin cloud cover (C_Thin) | cltcalipso_thin | % | (Time, lat, long) |
| CALIPSO thin cloud altitude (Z_T_Thin) | cltcalipso_thin_z | km | (Time, lat, long) |
| CALIPSO thin cloud emissivity (E_Thin) | cltcalipso_thin_emis | 1 | (Time, lat, long) |
| Surface elevation | SE | km | (Time, lat, long) |

*Data availability.* The monthly gridded dataset of LWCRE–LIDAR–Ed1 is available for the 2008–2020 time period at https://doi.org/10.14768/70d5f4b5-e740-4d4c-b1ec-f6459f7e5563 (Arouf et al., 2022). The data included in the dataset are presented in Table C1.

*Author contributions.* Conceptualization, investigation, and methodology were done by HC, AA, TVdG. Development was done by AA, TVdG with technical support from AF, RG, PR. Writing the original draft was done by AA, HC. All the authors brought contributions to the validation of the product and editing and writing the manuscript. The review was done by HC, AA. TS7

*Competing interests.* The contact author has declared that neither they nor their co-authors have any competing interests.

*Acknowledgements.* We thank Jean Lac for technical support and Erik Hojgard-Olsen for editing the text. We thank NASA/CNES for the CALIPSO level-1 data and the Mesocentre ESPRI/IPSL for the computational resources. We recognize the support of CNES, who supported the development of the CALIPSO–GOCCP product. The authors thank the anonymous reviewers for their useful comments and help in improving the paper. TS8

*Financial support.* This research has been supported by EADS (PhD grant for Assia Arouf) and the National Science Foundation (grant nos. PLR-1314156 and OPP-1801477 for Matthew D. Shupe and Michael R. Gallagher). TS9

*Review statement.* This paper was edited by Manfred Wendisch and reviewed by Hartwig Deneke and two anonymous referees.

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

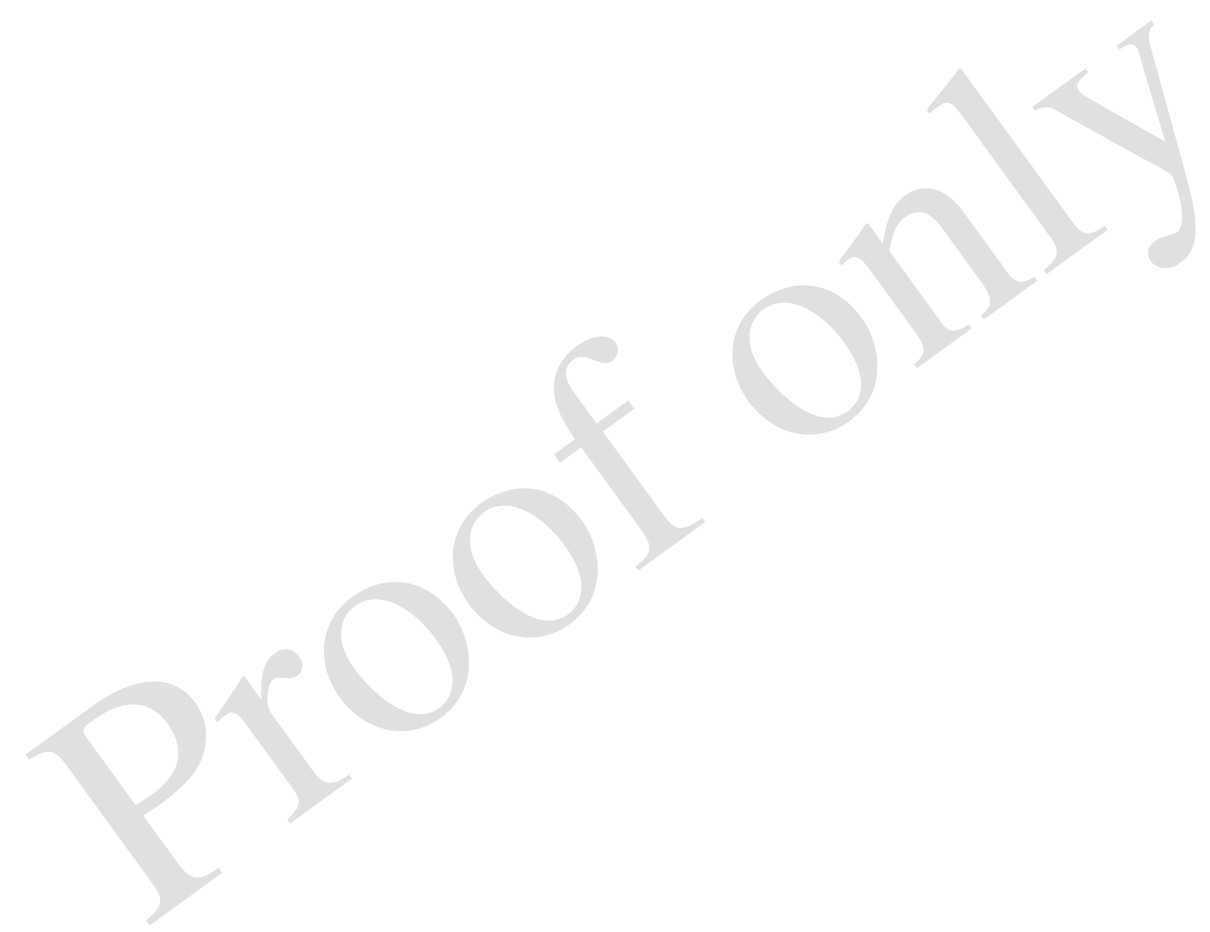

**Remarks from the typesetter**