# Peer review of "The Surface Longwave Cloud Radiative Effect derived from Space Lidar Observations"

_Atmospheric Measurement Techniques, 2021_

## Author Comment (AC1)

Hereafter, the reviewer comments are written in Black and the Answers to reviewer comments in Blue.

The study in general is written in very good English language. It is in parts a bit lengthy and I propose some cuts to make it better readable.
I have two main remarks that could be answered by additional discussion, as well as quite a number of specific remarks.

**Main remarks**
- For the LW CRE at the surface, it is crucial to estimate the cloud-base height correctly. The authors do not dwell on this problem very much, they basically just use what is readily available. There are, however, several approaches to retrieve it. One such approach uses CALIPSO (Mülmenstädt et al. 2018, doi: 10.5194/essd-10-2279-2018)

We thank the reviewer for pointing us to Mülmenstädt et al. 2018.
In the new version of the paper, we have computed the CALIPSO-GOCCP Surface LW CRE using the cloud-base height (called CBASE dataset) described in Mülmenstädt et al. 2018 in replacement of Z_fully-attenuated (Z_FA). Comparing the two estimates of the Surface LW CRE allows us to estimate the impact of using a more advanced cloud base estimate than  Z_fully-attenuated from lidar observations on CALIPSO-GOCCP CRE retrieval.

In the CBASE dataset, the Cloud base-height value is given at a horizontal resolution of 40km along the CALIPSO orbit track in the portion of the orbit where clouds are opaque. Along each CALIPSO orbit, we collocated the cloud-base height dataset with the GOCCP dataset whose horizontal resolution is 1/3 km along track. Over each 40 km orbit containing opaque cloud profiles, we replaced Z_fully-attenuated by the Cloud-base-height value given in the CBASE dataset, and then we computed Z_T_Opaque and the Surface LW CRE.

Figure 1 shows that CBASE values are distributed in all latitudes and are available in about one third of all the CALIPSO opaque profiles. This is because CBASE can only be retrieved when thin clouds are detected within the 40 km orbit piece that also contains opaque clouds profiles. Comparing Fig. 2a and Fig. 2b indicates that the subsample of opaque CALIPSO profiles where CBASE is documented contains both large values of CRE associated to mid and low level clouds located at mid-latitudes (upper right data in plot b) and small values of CRE (lower left), but it does not include the data where 2BFLX is much larger than CALIPSO-GOCCP which correspond to mid-latitude oceanic opaque clouds. When replacing Z_FA (Fig. 2b) by CBASE (Fig. 2c) in the CALIPSO-GOCCP algorithm, the CRE CALIPSO-GOCCP rises slightly almost everywhere because CBASE is lower in altitude than Z_FA, and CALIPSO-GOCCP CRE values lower than 18 W m$^{-2}$ are no more present. The latter correspond to both deep convective clouds and shallow boundary layer clouds (as discussed in Sect.7.1). The correlation between 2BFLX and CALIPSO-GOCCP is similar whether we use Z_FA (0.79) or CBASE (0.78) in the CALIPSO-GOCCP algorithm.
This sensitivity study suggests that using a more advanced cloud base height (here CBASE) estimate than Z_FA in CALIPSO-GOCCP algorithm will increase the CRE value retrieved in some opaque cloud profiles slightly, but it does not fundamentally change the results.
These new results are presented in Sect. 8.1.

[Figure]

[Figure]

**Figure 1:** Maps of the number of opaque clouds collocated with 2BFLX over oceans in February 2008 a) in all the CALIPSO-GOCCP collocated dataset b) same as a) but only where CBASE data are available.

[Figure]

**Figure 2:** Surface LW CRE derived from CALIPSO-GOCCP as a function of the one derived from 2BFLX. a) CALIPSO-GOCCP surface LW CRE (y axis) computed using the altitude of full lidar attenuation b) same as (a) but containing only the sub-sample of CALIPSO profiles where cloud base-height values are available from Mülmenstädt et al. 2018 c) same as b) but the CALIPSO-GOCCP surface LW CRE is computed using the cloud base-height values from Mülmenstädt et al. 2018 instead of the altitude of lidar full attenuation. The color scale indicates the number of occurrences at 5km resolution (footprint scale of CloudSat) over ocean in February 2008.

There is no discussion of the reasons for the linear relationship between $Z_{T,Opaque}$ and CRE. Can the authors produce some arguments on why this is the case, e.g. in the sense of what Corti and Peter (ACP 2009) did for TOA CRE?

This linear relationship is an empirical relation derived from radiative transfer calculations and verified in the observation at the TOA (Vaillant de Guélis et al., 2017, 2018). Corti and Peter (ACP 2009) also derived an empirical relationship (power laws) from radiative transfer computation. Our linear relationship can be considered as an approximation of the Corti and Peter power law.
We have added this information in the text of the manuscript Sect. 4.1.

**Specific remarks**

**C0)** l29 "captures"

Done

**C1)** l66 "ideal" and "everywhere" are a bit overdoing the statement. For the "everywhere" in particular, current lidar and radar are questionable due to the lack of swath.

"ideal way"has been replaced by a "possible way", and the word "everywhere" has been removed.

**C2)** l108 the optical depth is not measured but retrieved
"measurement" has been replaced by "retrieval"

**C3)** l200 "does not contribute" should be revised to be more quantitative. Why would an optical thickness of 5 in the green be exactly the threshold below which also no infrared radiation escapes the cloud?

We have removed previous Fig. 3 and therefore we have removed this sentence.

**C4)** l210 Well, CloudSat is also not optimal to detect cloud base in particular if it is liquid-water cloud

Right, we have added this information in the sentence.

**C5)** l212 What could be cases in which Z_T_Opaque is a better estimator of Z_Base than Z_FA? Z_FA is below Z_T_Opaque so it should always be better, **or am I mistaken**?

It is right that Z_FA is always a better estimator of Z_Base than Z_T_Opaque, but the surface LW CRE is not always driven only by Z_Base.
The Surface LW CRE is driven only by Z_Base in specific cases where enough condensed water (corresponding to an optical depth of 1) is confined within a single geometrically very thin cloud layer located exactly at Z_Base.  In all other cases, the cloud layers located above Z_Base up to the level where the cloud emissivity equal to 1 contribute to the surface LW CRE. This is why we cannot always use Z_FA and need to use Z_T_Opaque in numerous cases.
We have explained that more clearly in Section 3.1.2.

**C6)** l217 Why is this simpler? And what was the difference between the two choices?
It is simpler to write the equations with a single variable called "Z_T_Opaque". In the radiative transfer computation, we found that the Surface LW CRE depends linearly on the average altitude of the opaque cloud. We could have written the equations with a variable called "Z_mean_opaque" but that would have added another variable. Since we found in the observations that "Z_T_Opaque" is a good approximation of the averaged altitude of the opaque cloud in numerous cases, we chose to write the equation directly with "Z_T_Opaque".
We reverse the order of the sentences in the manuscript to clarify.

**C7)** l244 This is unclear. Is CRE computed at some aggregate scale? Because for each satellite footprint, there is either an opaque or thin cloud, and the CRE for the other type is zero. If it is aggregated in space or time, this should be stated here.
Right, this equation is only true for the gridded product.
For each satellite footprint, there is either an opaque or thin cloud, and the CRE for the other type is zero. For the gridded product, and at each grid point, the opaque surface LW CRE is weighted by the opaque cloud fraction, the thin surface LW CRE is weighted by the thin cloud fraction, and the total gridded surface LW CRE is the sum of the two.
We have re-written Section 3.4  for clarity.

**C8)** l269 Really theoretical or rather empirical?

We replaced "theoretical expressions" by "parameterization"

**C9)** l373 This point of missed multi-layer situations seems important enough to merit a broader discussion. How often may such systematic mistakes by CALIPSO-GOCCP occur?

We used the CBASE dataset by Mülmenstädt et al. (2018) to broadly discuss the impact of CALIPSO not seeing Z_Base in certain cloud situations. The details are given in the response to the main reviewers' comment here above and are reported in the manuscript in Sect. 7.
The comparison between CALIPSO-GOCCP and 2BFLX suggests that CALIPSO-GOCCP CRE is more frequently biased in the extra-tropical oceanic storm tracks than elsewhere, if 2BFLX is reliable there.

**C10)** l401 This idea can be tested, by comparing the humidity profiles used in the retrieval with the ones for these particular cases (e.g. from the reanalysis).

Right, we have tested this using re-analysis and added the results of the test in Sect. 7.

**C11)** l460 The authors provide the spatial scale, but should also note what is the temporal averaging. Or are these instantaneous values at time of satellite overpass?
Yes these are instantaneous values at the time of satellite overpass.

**C12)** l470 "months"
Done

**C13)** l476 Besides the biases, it would be useful to also report the other scalar quality indicators, RMSE and correlation coefficients, perhaps in a table.
The RMSE and correlation coefficients have been added in Table 1.

**C14)** l486 Is there any reference or evidence to substantiate this claim?

We have removed this sentence

**C15)** l498 How is this possible? The idea that it is due to humidity profiles is not plausible, since the same humidity profiles are used in both retrievals.

We have modified this section and included a new discussion based on new results in Sect. 7.

**C16)** l527 This section in my opinion is not very instructive, and there is room for making the paper more concise by dropping Fig. 15 and the corresponding text.

We agree with the reviewer and have removed this Fig. 15 and the corresponding text (former Sect. 7.1.2)

**C17)** l551 "suggest"
This section has been removed (previous comment)

**C18)** l570 before it was noted only for a specific location
Right, we now specify that in the text

**C19)** l597 It is overstated to say that CALIPSO "measures well" opacity – there are only two coarse classes distinguished.
Right, we replaced "are quite well measured" by "are documented"

**C20)** l830 I find the right-hand-side panel (Opaque cloud) a bit misleading, as the altitude Z_FA is just slightly above Z_Base. However, it often is quite near Z_Top, since it is at 5 optical depths (line 95).

The figure below shows the distribution of CBASE as a function of the distribution of Z_FA where CBASE data exists (about 30% of the CALIPSO-GOCCP opaque clouds profiles). It shows that 55.8% of the Z_FA values have less than 1km difference with CBASE and that the difference between Z_FA and CBASE can reach 12km.

[Figure]

**Figure 3:** CBASE as a function of Z_FA sample a) where CBASE is available in the collocated data used in Fig.2 b) same as (a) but when the difference between Z_FA and CBASE is less than 1 km. The color scale indicates the number of occurrences at 5km resolution (footprint scale of CloudSat) over ocean in February 2008.

**C21)** l840 To me this sketch isn't very helpful. I don't quite get the "actual" in the rectangular clouds in (A) and (B), from the caption I would rather understand, these are the fluxes computed by the radiative transfer modelling. Also, I don't see why the powerpoint-cloud-shaped-clouds and the rectangular carry different information with respect to the radiation arrows. Why do the arrows in (C) from the rectangular cloud end above the surface?

Right, we have removed this figure (previous Fig. 3)

**C22)** l898 I propose a different colour scale that does not suggest a division into two subsets.
Done

**C23)** l929 The mean biases lack units.
The unit is reported in the y-axis

**Reference :**

Corti, T. and Peter, T.: A simple model for cloud radiative forcing, 8, 2009.

Hinkelman, L. M. and Marchand, R.: Evaluation of CERES and CloudSat Surface Radiative Fluxes Over Macquarie Island, the Southern Ocean, Earth and Space Science, 7, https://doi.org/10.1029/2020EA001224, 2020.

Mülmenstädt, J., Sourdeval, O., Henderson, D. S., L'Ecuyer, T. S., Unglaub, C., Jungandreas, L., Böhm, C., Russell, L. M., and Quaas, J.: Using CALIOP to estimate cloud-field base height and its uncertainty: the Cloud Base Altitude Spatial Extrapolator (CBASE) algorithm and dataset (1), https://doi.org/10.1594/WDCC/CBASE, 2018.

Noel, V., Chepfer, H., Chiriaco, M., and Yorks, J.: The diurnal cycle of cloud profiles over land and ocean between 51° S and 51° N, seen by the CATS spaceborne lidar from the International Space Station, Atmos. Chem. Phys., 18, 9457–9473, https://doi.org/10.5194/acp-18-9457-2018, 2018.

Rutan, D. A., Kato, S., Doelling, D. R., Rose, F. G., Nguyen, L. T., Caldwell, T. E., and Loeb, N. G.: CERES Synoptic Product: Methodology and Validation of Surface Radiant Flux, 32, 1121–1143, https://doi.org/10.1175/JTECH-D-14-00165.1, 2015.

---

## Author Comment (AC2)

**Answers to Reviewer 2 comments**

Hereafter, the reviewer comments are written in Black and the Answers to reviewer comments in Blue.

**Methodology:**
Today's modern radiative transfer codes are very fast and quite accurate in the LW. In fact, they are often used to calculate LW CREs instantaneously in global products.
Additionally, it is now common for satellite retrievals of surface flux to account for subdaily variations in temperature/humidity and capture regional variations (e.g., east vs west tropical Pacific), climate events (e.g., ENSO) and extreme changes over polar regions. In this paper, LW surface CREs are determined using a simple parameterization approach that relates pre-calculated CRE values against only 5 CALIPSO cloud properties. Temperature/humidity profiles are prescribed using climatological means from ERA-Interim. The authors do not explain why such a simple approach is used when they could have used advanced radiative transfer codes to produce a more accurate product and account for temperature/humidity variations.

We used this simple approach because in future work we want to decompose temporal variations of the surface LW CRE into components in order to identify which cloud property that has driven the variations in surface LW CRE over the past 15 years. This decomposition can be done more easily and is more easily physically analyzed when using a simple parameterization approach, as shown in previous work at the TOA (eg. Vaillant de Guélis et al., 2017b, 2018).
In the first version of the manuscript, this explanation was given in the last paragraph of the Conclusion as a perspective.
Following the reviewer comment, we have now added this explanation in the Method Section, Sect. 3.1.

**Uncertainties:**
Uncertainties associated with the use of climatological monthly temperature/humidity profiles every 2 deg in latitude (with no dependence on longitude) are not adequately quantified. Only one very limited example (Fig. 5) is shown on the impact of humidity variability. The authors should provide a more comprehensive assessment. For example, how does their parameterization approach handle temperature/humidity variations over the maritime continent and subsidence regions in the same 2-deg latitude zone? In polar regions with a strong temperature inversion? Under different ENSO conditions (El Nino vs La Nina)? The use of climatological profiles likely leads to systematic errors that vary from region-to-region. These should be highlighted and discussed in the revised version of the paper.

The reviewer is right. It is now common for satellite retrievals of surface fluxes to account for sub-daily variations in temperature/humidity and to capture regional variations (e.g., eastern vs. western tropical Pacific Ocean), climate events (e.g., ENSO) and extreme changes over polar regions. As an example, CERES uses hourly reanalysis in Rutan et al., (2015) and 2BFLX uses 3-hourly atmospheric state variable data on a half-degree Cartesian latitude and longitude grid from AN-ECMWF.
Even though CALIPSO-GOCCP uses monthly mean temperature/humidity profiles (Figure 14a), monthly mean gridded surface LW CRE from CALIPSO-GOCCP are consistent with the monthly

2BFX gridded product based on sub-daily temporal resolution temperature/humidity profiles.. Nevertheless, instantaneous CALIPSO-GOCCP Surface LW CRE retrievals are likely more biased than the monthly mean gridded CALIPSO-GOCCP surface LW CRE, since it uses monthly mean temperature/humidity profiles. To estimate the error in the instantaneous CALIPSO-GOCCP surface LW CRE values, we compared the instantaneous CALIPSO-GOCCP values obtained using 6-hourly temperature/humidity profiles with the ones obtained using monthly means and with 2BFLX and CERES-CCCM, for one day at footprint scale.

We found that at CALIPSO footprint scale (1/3 km along-track), using sub-daily temperature/humidity profiles typically decreases the CALIPSO-GOCCP surface LW CRE down to 10 W/m2 in the extra-tropical clouds that warm the most (70 W/m2 instead of 80 W/m2).

Figure 1 shows (overall) that using sub-daily temperature/humidity profiles in CALIPSO-GOCCP leads to a lower agreement between CALIPSO-GOCCP and the other satellite products. This suggests that the differences between the three gridded daily products are likely due to other causes than CALIPSO-GOCCP using monthly mean temperature/humidity profiles.
Separate analyses of opaque and thin clouds (Fig. 2) show that for thin clouds, CALIPSO-GOCCP Surface LW CRE at CloudSat footprint scale (5km), retrieved using sub-daily temperature/humidity profiles, agree better with 2BFLX than CALIPSO-GOCCP Surface LW CRE retrieved using monthly mean profiles. In contrast, the agreement between GOCCP and other products are lower when using sub-daily temperature/humidity profiles in all other cases: opaque clouds and also thin clouds when compared to CERES-CCCM at SSF footprint (20km).

Following the reviewer comment, we have added a new section (Section 8.2) where we discuss these new results. We could use another radiative transfer code to produce a more accurate product that accounts for sub-daily temperature/humidity variations, as a nice perspective for a future version of this dataset. This would however be less helpful for decomposition of the Surface LW CRE into components.

[Figure]

Figure 1: Distribution of the Surface LW CRE: a) CALIPSO-GOCCP retrieved using monthly mean temperature/humidity profiles as a function of CERES-CCCM with data at 20 km resolution (CERES SSF footprint), b) same as (a) but CALIPSO-GOCCP retrieved using sub-daily temperature/humidity profiles, c,d) same as (a,b) but for 2BFLX instead of CERES-CCCM and using data at 5 km resolution (CloudSat resolution).

[Figure]

These details matter and should be explained.

We have added the following information in the text of the manuscript:
- a sentence in section 3.4 to explain how the 2x2 gridded product is created.
- a new Sect. 6.1 that describes how the comparisons with surface measurements are made.
- a sentence at the beginning of Sect. 7.1 to explain the collocation between the orbits.

2) The comparisons of CALIPSO-GOCCP with other data products summarized in Sections 6.1 and 6.2 are quite meaningful, as the authors collocated the data amongst CALIPSOGOCCP, CERES-CCCM and 2BFLX, thereby providing consistent sampling. One can draw clear conclusions about the differences shown, separate from sampling uncertainties.
These comparisons should be retained and even expanded upon.

We have expanded the comparison in quantifying the percentage that each packet of points represents in Fig. 13 and in adding two new Figures 16 and 17 of the same kind as Fig. 13.

3) The other comparisons are very difficult to interpret. The gridded monthly CERES EBAF surface fluxes are determined using full-swath CERES and MODIS data supplemented by geostationary imager measurements, thereby providing hourly global coverage. In contrast, CALIPSO-GOCCP only samples at the time of the satellite overpass and only over the satellite ground-track, resulting in far less spatial and temporal coverage.
Because of the differences in sampling, it is not obvious if these should agree even if both products were perfect. This makes it difficult to explain any differences between these products. As a result, the comparisons with CERES-EBAF should be removed from the paper as they add little value.

We agree that the difference between CERES-EBAF and other satellite products is also influenced by the full swath and the hourly coverage as listed by the reviewer. Moreover, it is also influenced by the difference in cloud detection between MODIS and the active remote sensing instruments.
We have removed CERES-EBAF from the paper as requested by the Reviewer.

In the revised version of the paper, we have added a new Section 6.4 on the diurnal cycle variation. In this new section, we first recall previous work (Noel et al., 2019 and Chepfer et al., 2019) based on CATS/ISS data that have shown that the average of CALIPSO daytime and nighttime observations is a good estimate of the daily mean over 55°S–55°N. We also present a new figure that compares the diurnal surface LW CRE variation observed over SIRTA ground-based sites and the CALIPSO-GOCCP surface LW CRE estimates. This comparison suggests that the absence of diurnal cycle correction might not be an important source of error in CALIPSO-GOCCP surface LW CRE estimates.
On this specific topic of the diurnal cycle, the new figure in the manuscript shows only the ground base site data and CALIPSO-GOCCP data, but not CERES-EBAF following the reviewer's request to remove CERES-EBAF from the paper. We have reported CERES-SYN on the figure below in the "response to reviewer document" because CERES-SYN is used in the generation of CERES-EBAF Surface fluxes. It suggests that diurnal cycle correction in CERES-SYN may introduce some bias in CERES-EBAF. Note that similar conclusion have been drawn by Hinkelman and Marchand (2020) over the Southern Ocean. For all seasons, they found SYN LW downwelling fluxes to compare well with the surface measurements between about 9 am and 3 pm, but poorly overnight (Fig. 6; 7 of Hinkelman and Marchand 2020). On the other hand, they also found the CloudSat downwelling fluxes to compare well with the surface measurements during both afternoon and night overpasses. They concluded that the correction in cloud diurnal cycle does not improve the CERES-EBAF restitution over this region and that, for all seasons, there is a systematical bias of CERES-EBAF of -11,6 W/m2 in the downwelling LW fluxes compared to their surface observations (HM, Fig. 16). Our results initial results (first version of the paper) indicate there is no difference between 2BFLX and CALIPSO in this specific region, while there is a difference of -10 W/m2 between CALIPSO-GOCCP and CERES-EBAF similar to the -11.6 W/m2 between CERES-EBAF and surface measurements.

F

[Figure]

Figure: Diurnal cycle of the Surface LW CRE a) ground base data and CERES-SYN data over SIRTA ground base site b) CERES-SYN data over at Summit c) ground base data over Summit. The stars correspond to CALIPSO-GOCCP data.

4) **Comparisons with ground measurements** : The methodology used to compare CALIPSO-GOCCP with ground measurements are not adequately explained. For example, are the surface measurements averaged in time or are instantaneous values used? How far from the ground site are the CALIPSO measurements? Are only coincident CALIPSO and ground site data used or are monthly means determined independently? If CALIPSO LW CREs were perfect, how closely should the LW CREs be to the ground-based measurements, given the sampling differences? Answering this last question will require some extra work. For example, one could envision using synthetic data with complete coverage and then compare that the CALIPSO sampling. Without such analyses, it is difficult to interpret the results.

Right, we have added a new Section 6.1 describing the methodology.

**Readability**:

The paper could use substantial restructuring, shortening and editing by the co-authors whose first language is English. Some suggestions are below but there are many more.

We have shortened the paper by removing previous Sect. 7.1.2 and previous Fig. 3
We have restructured the paper from Section 6 and afterwards based on Reviewer 2's comments,
We kept the sections before Section 6, since Reviewer 1 agreed with the structure of the paper.
Note that one co-author whose first language is English has spent substantial time in editing.

**Specific Comments**

**C1)** Line 17: delete "*long-term*". Thirteen years is short from a climate perspective.

Line 17 refers to space-based radiometers that have collected more than two decades of LW CRE, not thirteen years. To avoid misunderstanding we replaced long-term by more than two decades.

**C2)** Lines 18-20: "*The global surface LW CRE is estimated using long-term observations from space-based radiometers (2000–2021) but has some bias over continents and icy surfaces*."
This isn't entirely true from results shown in the paper. Over continents, diurnal cycles are quite pronounced, yet the CALIPSO-GOCCP method introduced in this paper does not resolve them. The impact of not sampling the maxima/minima in diurnal cycle heating on surface LW CRE is not discussed. The noisy surface LW CREs for CALIPSO-GOCCP in Fig. 17d is worrisome (by comparison, CERES-EBAF variations are smooth). Thus, to state categorically that CERES-EBAF is biased without pointing the obvious issues CALIPSO-GOCCP (Fig. 17d) is misleading. I suggest deleting this sentence.

This second sentence in the abstract was not intended to describe results of the current paper but the motivation for the current work. The statement "*has some bias over continents and icy surfaces*" is concluded from previous work such as Kay and l'Ecuyer (2013), that states "Liu et al. (2010) show that MODIS cloud detection retrievals perform better over the ocean than over ice and, as a consequence, MODIS data have unrealistically large increases in cloud amount from ice-covered to open water ocean surfaces"

To avoid misunderstanding, we have added "previous work have shown…" and we have re-written the sentence.

**C3)** Lines 20-21: "*To develop a more reliable **long time series** of surface LW CRE over continental and icy surfaces*".
The statement is not supported by results in the paper. As noted in the last comment, CALIPSO-GOCCP is very noisy over land and does not resolve the diurnal cycle. Also, thirteen years is not a long time series.

Following the reviewer comment, we replaced "long time series" by 13 years.

**C4)** Lines 22-25: "We show from 1D atmospheric column radiative transfer calculations, that surface LW CRE linearly decreases with increasing cloud altitude. These computations allow us to establish simple relationships between surface LW CRE, and five cloud properties that are well observed by the CALIPSO space-based lidar: opaque cloud cover and altitude, and thin cloud cover, altitude, and emissivity."
The authors are really using a simple parameterization to infer surface LW CRE. They should change: "These computations allow us to establish simple relationships..." to "These computations are used to develop **a simple parameterization for estimating surface** LW CRE from five cloud properties observed by the CALIPSO space-based lidar: etc."

Following the reviewer comment, we have replaced "relationships" by "parameterization"

**C5)** Lines 25-26: *"We use these relationships to retrieve the surface LW CRE at global scale over the 2008–2020 time period (27 Wm-2)."*
Is this sentence necessary? The one number doesn't say much on its own. Consider deleting this sentence.

Right, we have deleted this sentence

**C6)** Lines 40-41: *"defined as the change in the SW and LW radiation reaching the surface induced by the presence of clouds"*.
How is clear-sky defined? For example, by recalculating flux after removing clouds or by determining flux from cloud-free regions?

Clear sky is defined by recalculating fluxes after removing clouds with the same humidity and temperature profiles. This information has been added in the text.
Note that we do not add the information in the Introduction but later in the text, because it is linked to the discussion.

**C7)** Lines 59-65: As noted earlier, thirteen years is a relatively short record. While it may be tempting to think this record can be seamlessly continued with EarthCARE, as noted in the conclusions on line 617, there are some major hurdles to overcome: (i) EarthCARE will fly in a different orbit than CALIPSO (1400 h with 25-day repeat cycle), so that its groundtrack will not sample the same locations as CALIPSO; (ii) the EarthCARE lidar instrument characteristics (355-nm HSRL lidar) are very different from CALIPSO, so that their retrievals likely won't be consistent; (iii) it is unlikely CALIPSO and EarthCARE will overlap in time due to delays with the EarthCARE launch (currently March 2023). These issues will make it exceedingly difficult to construct a robust long-term CALIPSO- EarthCARE time series of surface LW CRE that is free of a discontinuity.
The authors should consider simply stating that the thirteen-year record can be useful for climate model evaluation and comparing against other satellite products without claiming this is a "*long-term*" record etc.

Following the reviewer comment, we removed the word "long-term" in lines 59-65.

**C8)** Line 98: Please define what a "gridbox" is. I believe it's a 2x2 deg latitude-longitude region?

Right we have added this information line 98.

**C9)** Line 120: "Figure 2 illustrates etc."
It appears there was a lot of smoothing and/or data gap filling used to produce Fig. 2. It would be far more informative to show these maps on a 2x2 deg latitude-longitude grid without any smoothing/gap filling to see how limited lidar sampling impacts the 13-year regional means.

Following the reviewer comment, we have re-plotted the figure without the smoothing.

**C10)** Sections 2.2.1 and 2.2.2: The descriptions of the CERES EBAF and 2BFLX products is inadequate. For example, one gets the impression that surface fluxes in CERES EBAF are inferred directly from CERES measurements, when in fact the main cloud property inputs to the radiative

transfer model calculations are from MODIS and geostationary imagers, atmospheric reanalysis, etc. **Two authors on this paper are responsible for those products. Surely, they can provide a better summary of their products**?

The summary of CERES-CCCM and 2BFLX have been re-written by the two authors.
CERES-EBAF has been removed from the paper as requested by the reviewer.

**C11)** Line 151-152: *"This product is sensitive to retrieval errors and biases introduced by the limited spatial and temporal characteristics of CloudSat and CALIPSO."*
This is also true of the CALIPSO-GOCCP product introduced in this paper, but appears to be significantly downplayed for some unknown reason.

Right, this is also true for CALIPSO-GOCCP, but our intent was not to downplay it.
We have completed the sentence to highlight that this is also the case for CALIPSO-GOCCP.

**C12)** Line 160: *"we selected three sites located in different regions"*.

Why only three sites? There are easily an order-of-magnitude more sites available (e.g., BSRN, SURFRAD, etc.). This is particularly surprising given the poor sampling obtained from CALIPSO.

The goal of the present study is to present for the first time a new product and to evaluate it against a set of other independent measurements. It is out of the scope of this paper to do an extensive comparison with all ground-based observations as in eg. Ruthan et al., (2015), or to do an extensive comparison with one specific satellite dataset, but that could be the dedicated purpose of future studies. We selected three ground-based sites where there is some specific science application for this product in the future and where we have some direct control of the data quality. Future work could include a more extensive comparison with more ground-base sites.

Sampling (or lack thereof) over these three sites is not adequately discussed. It appears from lines 179-180 that ground measurements at CALIPSO satellite overpass times are extracted and then averaged over each month. If there are days in which CALIPSO has no sampling over the site, are ground measurements from those days also excluded in the monthly means? How many CALIPSO samples per month are there over a typical site?
How is the spatial matching of CALIPSO and ground measurements determined? Are you using 2x2 deg latitude-longitude boxes centered on the ground site to match them (meaning there could be a >100-km separation between clouds from CALIPSO and the surface site)? Are the surface measurements averaged in time or are only instantaneous values used?

Right, we have added a new Section 6.1 describing the methodology.
We did not show all the details in the paper as this paper is not only dedicated to a comparison between the ground-based sites and the satellite data. Below is an example of analysis that we have done. It shows the difference between ground-based site data at the time of CALIPSO overpasses or averaged over 24h. We also performed analyzes in atmospheric circulation regimes as in Gallagher et al. 2020 (GRL) in order to group pixels and verify that the spatial differences between satellite grid box and the ground-base site does not impact the results over Greenland (not shown) The comparisons shown in the paper are robust to the different tests we have done.

[Figure]

[Figure]

**C13)** Similarly, when comparing surface measurements with CERES, are you including surface measurements over all 24-h of the day, since that is what CERES does? Are you ensuring the same days are sampled by CERES and the surface site in determining monthly mean differences? Is the CERES also aggregated over 2x2 deg regions like CALIPSO? There are so many unanswered questions…

As stated above, we have added Sect. 6.1

**C14)** Lines 166-167: *"Here, the clear sky flux is computed using a radiative transfer algorithm with measurements of temperature and humidity profiles"*.
How is all-sky determined? What instruments are used? What is the uncertainty in LW CRE from the ground site?

We have replaced the previous sentence by the following one: "..Here, the clear sky flux is computed using a radiative transfer algorithm with measurements of temperature and humidity profiles profiles (e.g., REFs), while the all sky flux is measured directly using a pair of upward-and downward-looking broadband pyrgeometers."

**C15)** Lines 171: "Here, the clear sky flux is computed from measurements of near surface temperature and vertical distribution of humidity".
Only near-surface temperatures? Why not use temperature/humidity profiles throughout the troposphere? Are they not available? If not, what uncertainty on LW CRE does this introduce?

At SIRTA, the clear sky flux is a parameterization made from the surface humidity, the integrated moisture content over the atmospheric column and the air temperature at 2 m. The details are given in Dupont et al. (2008) and this product has also been used in Rojas et al. 2021.
The resulting clear sky uncertainty is approximately +/-5 W/m2 as indicated in the data file.

**C16)** Lines 190-195: It appears temperature and humidity profiles in the parameterization are specified using climatology for every month of the year and at every 2 deg in latitude (see also Fig 6). Surprisingly, the parameterization does not appear to account for longitudinal variations in temperature and humidity, which can be very large (e.g., west tropical vs east tropical Pacific). In addition, year-to-year variability due to ENSO can result in very large variability in temperature and humidity. In polar regions, temperature inversions are frequent and moisture advection from lower latitudes can also cause large variations in temperature and humidity.

The paper notes, *"small variability of water vapor does not affect CRE very much compared to the fluxes themselves as the equivalent clear sky contribution is removed from CRE."* While this may be true for small variations, what about the actual variations that occur in nature? As noted above, the temperature and humidity variations can be quite pronounced.
To address the question of how humidity variations impact the parameterization, the reader is sent to entirely different sections of the paper (Section 4.1-4.3, yet only Section 4.2 is relevant). Only one simple example is provided (Fig. 5), in which we're told figure 5a uses a "*standard humidity profile*" and figure 5b uses an "*enhanced humidity profile*",
which are shown in Fig. A4. There is no explanation of where these profiles come from nor how representative they are of day-to-day or region-to-region variability in temperature and humidity. Nevertheless, they show a LW CRE difference of up to 7.7 Wm-2 just from a 10% change in humidity. This example does not address temperature profile differences since the standard and enhanced profiles are quite similar, as shown in Fig. A4b. This one example showing the sensitivity in CRE to temperature and humidity profile variations is insufficient.

We have added a new Section. 7b in the manuscript where we discuss this point.

**C17)** Lines 197-209: What is the main point that's being made here? Is it simply that space based lidars overestimate cloud base height and therefore underestimate surface LW CRE for opaque

clouds? If so, please state that up front. As it stands, it is unclear

The main point is not to state that "space-based lidars overestimate cloud base height and therefore underestimate surface LW CRE for opaque clouds". Rather, the point aims at explaining that:
1) in some specific opaque cloud cases, the Surface LW CRE is driven by the cloud base alone. This is when the cloud base level itself contains a lot of condensed water: enough to get an Emissivity close to 1 into the single cloud base layer, that is a geometrically thin layer <400m). If the space-based lidar overestimates the cloud base height, it therefore underestimates surface LW CRE for opaque clouds.
2) in all other opaque cloud cases, the Surface LW CRE depends on the vertical distribution of condensed water in the cloud, not only at the cloud base height. This is why we cannot always use Z_FA or Z_T_Opaque, as they depend on the vertical distribution of water within the opaque cloud.

We have removed the figure in this section, as it was not helpful and we have re-written the first part of the section to clarify.

**C18)** Line 209: "*To retrieve the surface LW CRE...*"
It may be helpful to insert "*from satellite*" after "LW CRE" since the previous paragraph discussed surface lidar.

We have removed this paragraph

**C19)** Section 3.2: Is this section necessary? It's obvious by the way CRE is calculated that the same surface temperature is used for clear and cloudy skies. That's all that can be done from satellite and is consistent with how GCMs calculate CRE. It is well known that CRE does not account for surface temperature changes due to the presence of the cloud.
Consider removing this section altogether or shortening it.

It is true that "that's all this can be done from satellites and that it is consistent with how GCMs calculate CRE", but CRE estimates from ground-based observations cannot keep surface temperature constant (eg. Shupe et al., 2013; Haeffelin et al., 2005). Since this paper compares CREs values derived from satellites and CRE values derived from ground-based sites, it is useful to remind the reader that their definitions cannot be exactly the same and therefore we do not expect these values to be exactly the same.

**C20)**  Line 250: "*we establish the relationship between*"
Consider revising to "*we derive a parameterization between*"

Done

**C21)** Line 253: So, the atmosphere only goes to 40 km? Please clarify.

The radiative transfer code simulates fluxes on 50 levels with a resolution of 1 km in the first 25 levels. We have changed the manuscript to clarify this.

**C22)** Lines 273-275: "We tested both Z_FA and Z_Topaque for estimating etc."

Couldn't the choice of whether to use Z_FA or Z_Topaque be better determined from radiative transfer model simulations instead of comparisons with other data products/ground measurements? Using other data sources introduces all kinds of additional issues, making the decision more complicated.

As explained in C17, the better choice between Z_FA and Z_Topaque depends on the vertical distribution of condensed water within the atmospheric column. Therefore, it is cloud scene dependent. As shown in the new Fig. 14b, most of the cloud scenes in Z_T_Opaque are more adapted but in some particular scenes Z_FA would be more adapted.

**C23)** Line 342: Section 5.2 Gridded Product

How is the gridded product determined? CALIPSO sampling is restricted to the satellite ground-track. Are the CALIPSO lidar measurements simply gridded and averaged into 2x2 deg latitude-longitude regions or is a different approach used? Is there a minimum number of CALISPO samples required in each 2x2 gridbox? Are monthly regional averages determined from daily means or are all CALIPSO samples in a month summed and divided by the total?
How the 2x2 gridded product is created ? Is there a minimum number of CALISPO samples required in each 2x2 gridbox?

The sampling, gridding, averaging of the CALIPSO-GOCCP cloud product algorithm is described in details in previous papers (eg. Chepfer et al. 2010/2011; Chepfer at al. 2013; Guzman et al. 2017, Cesana et al.,2012). The gridded CRE uses gridded CALIPSO-GOCCP as input in the same way as Vaillant-de-Guélis et al. (2017a).
This information has been added in Sect. 3.4.

**C24)** Line 343: Figure 9: Was any smoothing/gap filling used to create this figure? It would be far more informative to show the actual map with no smoothing/gap filling and for 2x2 deg latitude-longitude resolution.

Yes, there was some smoothing in the first version of this figure and we have redone the figure without smoothing in the new version. We also changed the color scale as suggested by Reviewer 1, so that it does not suggest a division into two subsets

**C25)** Line 372: Fig. B1 (bottom) compares CERES-EBAF against CALIPSO-GOCCP whereas Figure 10 compares CERES-CCCM. Is the label incorrect in Fig. B1?

Yes, the label was incorrect in Fig. 1B, it is CERES-CCCM. It has now been corrected.

**C26)** Line 378: I believe the CCCM approach uses full-resolution CALIPSO/CloudSat data along the ground-track over 20 km CERES footprints, so it should detect anything CALIPSO-GOCCP detects. There should be no "missing" clouds. Please clarify.

Right we have updated this sentence.

**C27)** Section 6.1: What conclusions can one make based upon these comparisons? It's not enough to just show the differences. Can one say that by not including CloudSat, CALIPSOGOCCP LW CREs

are biased low by 15-20 Wm-2 compared to CCCM and 2BFLX in regions of deep convection and stratocumulus?

One can say that by not including CloudSat, CALIPSO-GOCCP LW CREs are biased low by typically 10 Wm-2 compared to 2BFLX and 15 W/m2 compared to CCCM in regions of deep convection. In stratocumulus, CALIPSO-GOCCP is biases low by typically 5 Wm-2 compared to 2BFLX and -15 W/m2 compared to CCCM.
We have added this information at the end of Section 7.1 in the new version of the manuscript.

**C28)** Line 386: "CERES-CCCM (20 km footprint)". As noted above, CCCM uses full-resolution CALIPSO and CloudSat data but reports the results over 20-km CERES footprints. Please revise "20 km footprint" as it makes it sound like the full CALIPSO-CloudSat data are not used in CCCM, which is not the case.

We revised the text according to the reviewer comment.

**C29)** Line 433: "In global annual mean, CALIPSO–GOCCP is equal compared to CERES–EBAF and slightly higher compared to 2BFLX"
How do we know the consistency between CALIPSO-GOCCP and CERES-EBAF is for the right reasons given how different their time-space sampling is? This agreement could be for all the wrong reasons, which makes such comparisons of limited value.

As mentioned above, we have removed CERES-EBAF from this paper.
Nevertheless, please note that the first version of this manuscript did not say that "the consistency between CALIPSO-GOCCP is for the right reason". There are compensation errors in all satellite product inter-comparisons.

**C30)** Line 457: "Section 6.4 Comparison with ground-based stations at gridded scale"
 Lines 485-489: "*CALIPSO does not see the cloud base in many stratiform-type clouds, as an example, but this does not lead to as big of an issue in the surface LW CRE retrieval because the stratiform cloud base is not very far from the point of attenuation of the lidar.*"
This statement is inconsistent with the example shown in Fig. 10 (orbit C), which shows CALIPSO-GOCCP to be lower than CERES-CCCM by 15 Wm-2. Please clarify

We have removed this sentence.
In the revised version of the paper, we have added a new Sect. 8 where we discuss the impact of CALIPSO not seeing the cloud base.

**C31)** Lines 506-510: This argument acknowledges the challenges of comparing ground-based and satellite estimates but does not quantify the impact of these challenges. As a result, it is unclear what to conclude from those comparisons? If CALIPSO-GOCCP were perfect, how closely to the ground measurements should the LW CREs be, given the substantial sampling challenges? This is hard for the reader to know since the methodology for comparing the ground and satellite LW CREs was not provided.
The methodology of comparison between satellite product and ground-based site has been described in more detail in the new Sect. 6.1, included in the new version of the paper

**Reference :**

Corti, T. and Peter, T.: A simple model for cloud radiative forcing, 8, 2009.

Hinkelman, L. M. and Marchand, R.: Evaluation of CERES and CloudSat Surface Radiative Fluxes Over Macquarie Island, the Southern Ocean, Earth and Space Science, 7, https://doi.org/10.1029/2020EA001224, 2020.

Mülmenstädt, J., Sourdeval, O., Henderson, D. S., L'Ecuyer, T. S., Unglaub, C., Jungandreas, L., Böhm, C., Russell, L. M., and Quaas, J.: Using CALIOP to estimate cloud-field base height and its uncertainty: the Cloud Base Altitude Spatial Extrapolator (CBASE) algorithm and dataset (1), https://doi.org/10.1594/WDCC/CBASE, 2018.

Noel, V., Chepfer, H., Chiriaco, M., and Yorks, J.: The diurnal cycle of cloud profiles over land and ocean between 51° S and 51° N, seen by the CATS spaceborne lidar from the International Space Station, Atmos. Chem. Phys., 18, 9457–9473, https://doi.org/10.5194/acp-18-9457-2018, 2018.

Rutan, D. A., Kato, S., Doelling, D. R., Rose, F. G., Nguyen, L. T., Caldwell, T. E., and Loeb, N. G.: CERES Synoptic Product: Methodology and Validation of Surface Radiant Flux, 32, 1121–1143, https://doi.org/10.1175/JTECH-D-14-00165.1, 2015.

---

## Author Response (AR2)

Hereafter, the reviewer comments are written in Black and the Answers to reviewer comments in Blue.

Answers to Reviewer 3 comments

**1\*** the name "CALIPSO–GOCCP product": I find that using this term both for the actual cloud products (which has been published previously under this name) as well as the LW-CRE dataset obtained by the author's parametrization based on CALIPSO-GOCCP based cloud information is unfortunate and confusing. Please change this and use a distinct name/acronym for your LW CRE estimate.

We agree with the reviewer comment. The new product name is **"LWCRE-LIDAR-Ed1"** for **'LW Cloud Radiative Effect derieved from space Lidar observations Edition 1'** and the acronym is $CRE_{LIDAR}$. We have updated the manuscript using this new name

**2\*** Also, the article is rather unspecific about the form of publication. Which part of the data set will be available? Which variables will be included? Will the published dataset comprise the swath product, or only the gridded product. Given the overlap of authors with the creators of the GOCCP product, is this LW CRE product indeed intended to become part of the CALIPSO-GOCCP dataset? Please clarify this important aspect more thoroughly, ideally creating DOIs for the new data sets.

- Which part of the data set will be available?

We provide the monthly gridded product over the 2008-2020 time period.

- Which variables will be included? :

The table below gathers the variables that will be included in the dataset and provide their description. This Table is included in a new Appendix C named Data availability.

- Will the published dataset comprise the swath product, or only the gridded product.

For now we provide the 2×2 lat-lon gridded monthly product in a netcdf format over the 2008-2020 time period

(https://doi.org/10.14768/70d5f4b5-e740-4d4c-b1ec-f6459f7e5563) and we will provide the gridded daily product as well as the orbit instantaneous product soon.

- is this LW CRE product indeed intended to become part of the CALIPSO-GOCCP dataset?

Yes. As this surface LW CRE is derived form the CALIPSO-GOCCP-OPAQ cloud properties, it will become part of the CALIPSO-GOCCP-OPAQ dataset.

- creating DOIs for the new data sets.

According to the reviewer comment we have created the following DOI for this dataset: https://doi.org/10.14768/70d5f4b5-e740-4d4c-b1ec-f6459f7e5563 and have been added in the paper.

**Table 2: LWCRE-LIDAR-Ed1** Monthly Gridded Products: Definitions and Variable Names

| Geophysical Quantity | Variable name in the nc file | Unit | Dim |
|---|---|---|---|
| time | time | "month" | time |
| lon | Longitude | °E | lon |
| lat | Latitude | °N | lat |
| Surface Cloud Radiative Effects Net Longwave Flux Monthly Means (Surface LW CRE) | sfc_cre_net_lw_mon | W m$^{-2}$ | (time, lat, lon) |
| Surface Opaque Cloud Radiative Effects Net Longwave Flux Monthly Means (Surface LW Opaque CRE) | sfc_cre_net_lw_mon_opaque | W m$^{-2}$ | (time, lat, lon) |
| Surface Thin Cloud Radiative Effects Net Longwave Flux Monthly Means (Surface LW Thin CRE) | sfc_cre_net_lw_mon_thin | W m$^{-2}$ | (time, lat, lon) |
| Surface Opaque Cloud Radiative Effects Net Longwave Flux Monthly Means derived using Fully Attenuated Altitude (Surface LW Opaque CRE_Z_FA) | sfc_cre_net_lw_mon_Z_FA | W m$^{-2}$ | (time, lat, lon) |

| Top Of the Atmosphere Cloud Radiative Effects Longwave Flux Monthly Means (TOA LW CRE) | toa_cre_lw_mon | W m$^{-2}$ | (time, lat, lon) |
|---|---|---|---|
| Top Of the Atmosphere Opaque Cloud Radiative Effects Longwave Flux Monthly Means (TOA LW Opaque CRE) | toa_cre_lw_mon_opaque | W m$^{-2}$ | (time, lat, lon) |
| Top Of the Atmosphere Thin Cloud Radiative Effects Longwave Flux Monthly Means (TOA LW Thin CRE) | toa_cre_lw_mon_thin | W m$^{-2}$ | (time, lat, lon) |
| CALIPSO Opaque cloud cover (C_Opaque) | cltcalipso_opaque | % | (time, lat, lon) |
| CALIPSO Opaque cloud altitude (Z_T_Opaque) | cltcalipso_opaque_z | km | (time, lat, lon) |
| CALIPSO Fully Attenuated altitude (Z_FA) | zopaque | km | (time, lat, lon) |
| CALIPSO Thin cloud cover (C_Thin) | cltcalipso_thin | % | (time, lat, lon) |
| CALIPSO Thin cloud altitude (Z_T_Thin) | cltcalipso_thin_z | km | (time, lat, lon) |
| CALIPSO Thin cloud emissivity (E_Thin) | cltcalipso_thin_emis | 1 | (time, lat, lon) |
| Surface Elevation | SE | km | (time, lat, lon) |

**3\*** Referring to the following sentence in the conclusions: "All three satellite datasets exhibit some differences relative to ground-based measurements". This is a rather vague and unspecific comment and the abstract is rather more vague on this L26. Looking at Sec6.2, there seems to be a 10W/m2 bias in LW-CRE over polar regions! Is this difference significant / indicative of a general lack of understanding in the LW radiation budget? Is this discrepancy caused by clouds? I suggest that the authors to add some more concrete text on this both in the abstract and conclusions.

The 10 W m$^{-2}$ bias in LW-CRE over Summit in winter is partly due to CALIPSO-GOCCP missing thin cloud below 2 km above ground level in winter, as shown in Lacour et al. (2017).

This specific information was lacking in manuscript and has been added in the conclusion (L673) and in Sect. 6.2 (L432).

In the abstract we have now stated that this new product shows good correlations with other datasets (L26), because we do not want to focus only on Greenland Summit in winter.

We updated Sect. 2.2.2 to provide a more complete paragraph describing the 2BFLX product.

Toward that point, I also ask that the authors add the bias between obs and the satellite-based products in Table 1.

Done

Specific comments:

L22: "period, We" => use full-stop to separate sentence

Done

L283: "GOCCP observations" => change to "GOCCP product"

Done

Figs 3,4 and 6: please indicate units in linear relations / coefficients.

We have rewritten more properly the equation in the Figures,

The unit of linear coefficients is now given in the figure caption and in the text describing the Figures.